# Remove the Ambiguity: Few-shot Multimodal Anomaly Detection Using Crossmodal Feature Replacer

Yuan Guo[1]  Wanqi Zhang[1]  Xu Wang[1 2]

## Abstract

A key challenge in reconstruction-based multimodal anomaly detection is the one-to-many crossmodal mapping problem: a single 3D feature may correspond to multiple plausible RGB appearances, causing deterministic crossmodal regression to collapse valid targets into over-smoothed reconstructions and thereby weaken anomaly discrimination. In this paper, we propose *Crossmodal Feature Replacer* (CFR), a self-supervised framework that addresses this failure mode through selective inference-time feature replacement. CFR first learns bidirectional cyclic mappings for coarse crossmodal reconstruction, then identifies unreliable reconstructed features and selectively replaces them with high-confidence normal features to correct ambiguity-induced reconstruction failures. Extensive experiments on MVTec 3D-AD and Eyecandies under few-shot settings show that CFR consistently outperforms prior methods. In the challenging 1-shot setting, CFR achieves AUPRO scores of 92.3 and 82.7 at 30% FPR, together with image-level AUROC scores of 74.0 and 75.9, on MVTec 3D-AD and Eyecandies, respectively. Code is available at https://github.com/Yuan-Honoka-Guo/CFR.

## 1. Introduction

Anomaly detection is a core problem in industrial inspection, where defective samples are rare and expensive to annotate in advance. Consequently, most industrial anomaly detection methods are formulated in an unsupervised or self-

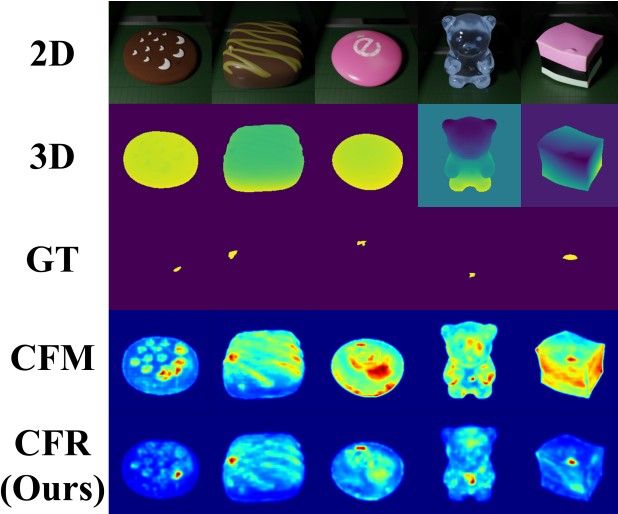

*Figure 1.* Qualitative examples on the Eyecandies benchmark illustrating ambiguity in RGB–3D anomaly detection. GT denotes the ground truth. The bottom row shows the segmentation results of CFR. By selectively correcting unreliable crossmodal reconstructions at inference time, CFR yields more accurate and discriminative anomaly segmentation. Colors ranging from blue to red indicate increasing anomaly scores.

supervised manner, learning normal patterns from defect-free data and detecting anomalies as deviations at test time. While many existing methods rely primarily on RGB images, RGB inspection can be unreliable under illumination variation or when defects manifest mainly as subtle geometric deviations. Combining RGB images with 3D point clouds provides complementary appearance and geometric cues, enabling more reliable detection of defects that may be ambiguous in a single modality. The release of multimodal benchmarks such as MVTec 3D-AD (Bergmann et al., 2022) and Eyecandies (Bonfiglioli et al., 2022) has further stimulated research on RGB–3D anomaly detection.

Existing multimodal anomaly detection methods can be broadly grouped into three paradigms. Teacher–student methods transfer normal knowledge across modalities through feature distillation, but are often limited by imperfect crossmodal supervision and relatively weak localization ability (Rudolph et al., 2023; Gu et al., 2024). Memory-based methods store multimodal normal features

[1]The College of Computer Science, Sichuan University, Chengdu, China [2]Centre for Frontier AI Research (CFAR), Agency for Science, Technology and Research (A*STAR), Singapore. Correspondence to: Xu Wang <wangxu.scu@gmail.com>.

*Proceedings of the 43[rd] International Conference on Machine Learning*, Seoul, South Korea. PMLR 306, 2026. Copyright 2026 by the author(s).

for retrieval-based scoring, but typically incur substantial memory overhead and face scalability challenges in high-resolution or dense settings (Chu et al., 2023; Wang et al., 2023b; Cao et al., 2024a; Wang et al., 2024). Reconstruction-based methods instead learn crossmodal mappings to reconstruct one modality from the other and detect anomalies through reconstruction errors (Chen et al., 2023; Bi et al., 2024; Zavrtanik et al., 2024b). This paradigm is particularly appealing because it directly exploits crossmodal inconsistency for anomaly detection. However, most existing reconstruction-based approaches rely on deterministic crossmodal regression between heterogeneous modalities, making them brittle precisely when crossmodal correspondence is one-to-many.

In practice, the mapping between RGB and 3D features is often ambiguous: a feature in one modality may correspond to multiple equally plausible features in the other. For example, objects with nearly identical geometry may still exhibit markedly different RGB appearances, especially around color transitions and appearance boundaries. Under such ambiguity, deterministic crossmodal regression tends to collapse multiple valid targets into an averaged solution, producing over-smoothed reconstructions rather than preserving a faithful correspondence. This failure mode weakens the discriminative power of reconstruction residuals and makes the model sensitive to benign appearance variation in normal samples. Consequently, large reconstruction errors may arise even in normal but visually complex regions, leading to spurious anomaly responses near appearance boundaries, as illustrated in Figure 1.

To address this limitation, we propose *Crossmodal Feature Replacer* (CFR), a self-supervised framework that resolves this failure mode through selective inference-time feature replacement. CFR first learns bidirectional cyclic mappings to obtain coarse crossmodal reconstructions and stabilize feature alignment between RGB and 3D modalities. CFR then detects unreliable reconstructed features, namely those affected by ambiguity-induced collapse. Similar strategies for disambiguating crossmodal correspondences have been explored in crossmodal retrieval, such as semantic-consistent contrastive hashing (Peng et al., 2026) and neighbor-aware disambiguation (Su et al., 2026), as well as partial-label alignment methods (Su et al., 2025; Liu et al., 2024). Instead of relying on retrieval to explain the entire input, CFR invokes retrieval only for these unreliable regions and selectively corrects them with high-confidence normal features. In this way, CFR does not force a single deterministic reconstruction to explain all plausible crossmodal correspondences. Rather, reconstruction provides a coarse crossmodal prediction, while retrieval serves as a targeted correction mechanism when reconstruction becomes unreliable. This coarse-to-fine design directly targets ambiguity-induced reconstruction failures and is particu-

larly effective under few-shot settings, where normal data are scarce and reconstruction ambiguity is harder to resolve.

Our contributions are summarized as follows:

- We identify the one-to-many crossmodal mapping problem as a key failure mode in multimodal anomaly detection and show that deterministic crossmodal regression can collapse multiple valid targets into over-smoothed reconstructions, weakening anomaly discrimination under few-shot settings.
- We propose *Crossmodal Feature Replacer* (CFR), a multimodal anomaly detection framework that addresses this failure mode through selective inference-time feature replacement.
- We develop a coarse-to-fine correction mechanism in which reconstruction provides a coarse crossmodal prediction and retrieval is invoked only to correct unreliable reconstructed features, reducing ambiguity-induced reconstruction failures without resorting to wholesale retrieval-based replacement.
- Extensive experiments on MVTec 3D-AD and Eyecandies show that CFR consistently outperforms prior methods under few-shot regimes, demonstrating its effectiveness in data-scarce industrial scenarios.

## 2. Related Work

**Reconstruction Methods.** Reconstruction-based anomaly detection is a fundamental paradigm for unsupervised defect detection, where anomalies are identified through reconstruction discrepancies. Early methods mainly operate in the pixel space, leveraging autoencoders or GANs to model normal data distributions (Akcay et al., 2018; Bergmann et al., 2019; Akçay et al., 2019; Gong et al., 2019; Schlegl et al., 2019). However, pixel-level reconstruction is often sensitive to noise and prone to over-generalization, motivating more robust strategies such as masked inpainting and discriminative restoration (Zavrtanik et al., 2021a;b). To better capture higher-level semantics, recent studies have shifted toward feature-space reconstruction, where deep representations extracted from pretrained networks are reconstructed for anomaly scoring (Shi et al., 2021; Schlüter et al., 2022). Hybrid approaches that combine pixel- and feature-level cues have also been explored to enhance localization performance (Hou et al., 2021; Zhang et al., 2024). More recently, diffusion-based generative models have emerged as an alternative reconstruction framework by modeling normality through iterative denoising processes (Zhang et al., 2023; Mousakhan et al., 2024). In multimodal settings, reconstruction and correspondence learning across RGB images and 3D point clouds further enable exploiting complementary appearance and geometric information for more reliable defect detection (Chen et al., 2023; Bi et al., 2024; Zavrtanik

et al., 2024a). CFM (Costanzino et al., 2024) learns crossmodal feature mappings from normal samples and detects anomalies through prediction inconsistencies between observed and reconstructed representations. However, since CFM relies on learning a single deterministic mapping between heterogeneous modalities, it may struggle in scenarios where one-to-many crossmodal correspondences exist, limiting its robustness under large appearance diversity.

**Memory Bank.** Memory-based methods have become an important paradigm in unsupervised anomaly detection, as they explicitly model normality through feature storage and retrieval rather than reconstruction alone. Early works incorporate external memory modules into autoencoders, where a set of normal prototypes is retrieved to alleviate the over-generalization issue of conventional reconstruction models (Gong et al., 2019; Park et al., 2020). More recent retrieval-based approaches represent normality using pretrained feature embeddings and detect anomalies via deviations from stored representations, such as SPADE and PaDiM (Cohen & Hoshen, 2020; Defard et al., 2021). PatchCore further constructs a patch-level memory bank and performs nearest-neighbor retrieval for accurate anomaly localization (Roth et al., 2022), while EfficientAD improves efficiency by avoiding expensive exhaustive search in large memory banks (Batzner et al., 2024). In multimodal anomaly detection, memory-based designs are especially useful for capturing complementary normal patterns across different modalities (Chu et al., 2023; Wang et al., 2023b; Cao et al., 2024a; Tu et al., 2024). However, memory-based anomaly detection methods often incur high computational overhead due to nearest-neighbor retrieval, and their performance can degrade under representation noise and distribution shifts, limiting generalization in real-world scenarios.

Motivated by these limitations, we focus on addressing unreliable crossmodal reconstructions that arise from the one-to-many mapping ambiguity. Rather than relying on explicit memory bank matching with limited generalization, CFR introduces a crossmodal retrieval and feature replacement mechanism that explicitly targets ambiguous reconstructed features. Specifically, CFR identifies low-confidence reconstructed features and selectively replaces them with attention-weighted high-confidence candidates retrieved from memory banks, yielding a more faithful and discriminative representation. Similar ideas for robust crossmodal alignment and few-shot scenarios have been investigated in the literature, including robust domain alignment (Yin et al., 2025), correspondence-free alignment (Wang et al., 2023a), and few-shot pseudo-label or meta-learning strategies (Li et al., 2026; Lei et al., 2022), and methods for handling sparse or biased supervision in label distribution learning (Kou et al., 2024; 2025).

## 3. Methodology

We propose CFR, a structured multimodal anomaly detection framework that combines reconstruction and feature retrieval. CFR enforces feature fidelity while benefiting from the generalization ability of neural networks in industrial scenes where training samples are scarce. CFR improves crossmodal feature consistency by combining coarse reconstruction with selective retrieval-based correction.

Specifically, CFR operationalizes this synergy through three iterative stages (as shown in Figure 2): Crossmodal Feature Learning, Crossmodal Feature Retrieval, and Feature Replacement. The first stage learns a cycle-consistent crossmodal alignment between RGB and 3D features while simultaneously constructing clustered feature representations for both modalities. Then, with the features obtained, we train a simple attention network for the feature retrieval task. The third stage takes place at inference time, using an iterative approach where 1) features in RGB and 3D representation are reconstructed, 2) reconstructed RGB features are filtered with memory bank to mark out unreliable ones, 3) the retrieval network is utilized to search for more reliable features to replace those that cause inconsistencies.

In this way, CFR can significantly mitigate the one-to-many mapping collapse problem in training a reconstruction network while simultaneously maintaining the ability to generalize to unseen samples under few-shot settings. First, CFR reconstructs crossmodal features that encode the geometry–color correspondence observed in normal samples, a correspondence that breaks down in the presence of anomalies. Second, for reconstructed features, the replacement module serves as a critical confidence scanner and a fallback mechanism for false anomalous features.

### 3.1. Crossmodal Feature Learning

**Feature Extraction.** The initial step of CFR is feature extraction, which obtains RGB features and 3D point cloud features using networks pretrained on ImageNet and PointNet as feature extractors. The pretrained networks are kept frozen, and layer pruning is applied as performed in previous works (Costanzino et al., 2024) for computational efficiency.

For a $H \times W \times C$ image, we use the middle layer output of the feature extractor of shape $H_p \times W_p \times D_{2D}$ and interpolate it to an image resolution feature map as $E_{2D}$. Using the feature extractor for the 3D point cloud of shape $N \times 3$, we encode a subset of 3D points and interpolate their features to all points using a weighted sum of nearest centers, following (Wang et al., 2023b). According to the standard setting, we use pixel-registered 3D datasets (Bergmann et al., 2022; Bonfiglioli et al., 2022), which means that every point feature $F_{3D}$ in $E_{3D}$ can now be matched with a pixel feature

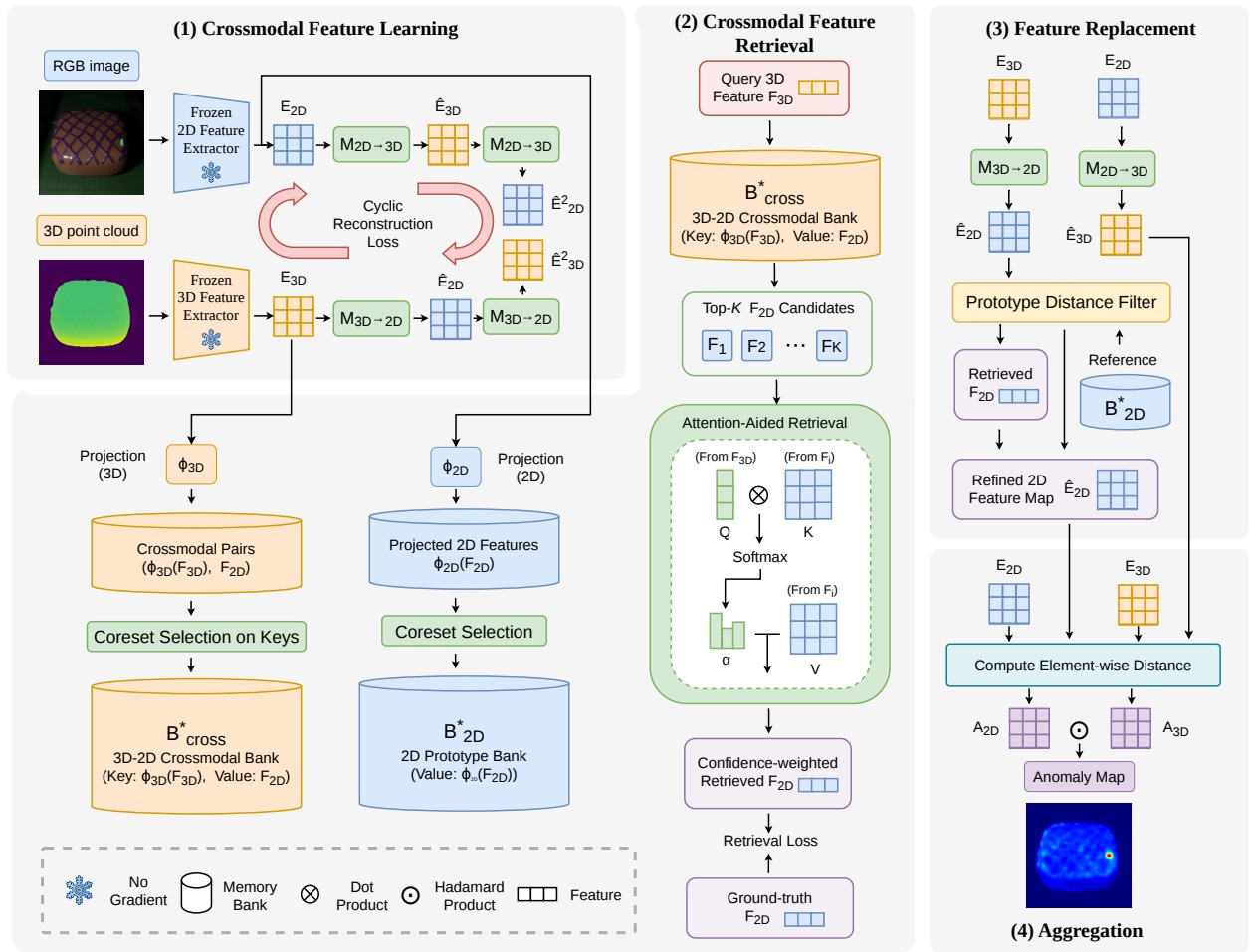

*Figure 2.* Overview of *Crossmodal Feature Replacer (CFR)*. (1) Crossmodal Feature Learning trains modality-to-modality mappings to reconstruct feature maps across RGB and 3D modalities. (2) Crossmodal Feature Retrieval constructs feature prototypes and retrieves confident 2D feature candidates from the crossmodal memory bank. (3) Feature Replacement identifies unreliable reconstructed features with a prototype distance filter and replaces them with reliable retrieved features. (4) Aggregation combines the reconstructed feature maps to produce the final anomaly score map.

$F_{2D}$ in $E_{2D}$.

**Cyclic Feature Mapping.** As we have obtained $E_{2D}$ and $E_{3D}$, we employ a pair of Cyclic Feature Mapping functions denoted by $M_{2D \to 3D}$ and $M_{3D \to 2D}$ to perform the reconstruction of features in one modality using the features of the other modality. Specifically, given the 2D and 3D feature maps $E_{2D}$ and $E_{3D}$, we have

$$\hat{E}_{3D}^1 = M_{2D \to 3D}(E_{2D}), \quad \hat{E}_{2D}^1 = M_{3D \to 2D}(E_{3D}). \quad (1)$$

This produces reconstructed 2D and 3D feature maps, respectively, and it requires the mapping functions to have injectivity for successfully mapping to the corresponding features in the other modality. Since neural networks project representations across heterogeneous feature spaces, their optimization can benefit from regularization constraints that encourage approximately invertible crossmodal correspon-

dence. Concretely, we want to apply the above mapping continuously and obtain

$$\hat{E}_{3D}^2 = M_{2D \to 3D}(\hat{E}_{2D}^1), \quad \hat{E}_{2D}^2 = M_{3D \to 2D}(\hat{E}_{3D}^1). \quad (2)$$

The superscript indicates how many times the initial feature is mapped. At training time, we optimize all feature samples in the reconstruction cycle using cosine distance as a reconstruction fidelity metric and train $M_{2D \to 3D}$ and $M_{3D \to 2D}$ jointly. Thus, the cyclic reconstruction loss is the aggregation of

$$\mathcal{L} = \sum_{i \in \{1,2\}} \left( 1 - \frac{E_{2D} \cdot \hat{E}_{2D}^i}{\|E_{2D}\| \, \|\hat{E}_{2D}^i\|} + 1 - \frac{E_{3D} \cdot \hat{E}_{3D}^i}{\|E_{3D}\| \, \|\hat{E}_{3D}^i\|} \right). \quad (3)$$

It is important to note that, due to the variability of the nominal samples, the mapping between 2D and 3D may

not be unique (Costanzino et al., 2024). As shown in the rightmost column of Figure 1, different licorice sandwiches can have different colors for every layer while having shapes that are approximately the same.

### 3.2. Crossmodal Feature Retrieval

**Build Crossmodal Banks.** Along with training cyclic feature mapping, we also utilize the encountered $F_{3D}$ and $F_{2D}$ pairs to construct memory banks. We propose a crossmodal memory bank that does not store features of a single modality but uses features of one modality as the index key for the other, enabling a key-value search paradigm. To reduce the search time, we employ two random linear projections, $\phi_{2D}$ and $\phi_{3D}$, based on the Johnson–Lindenstrauss theorem (Dasgupta & Gupta, 2003), to map RGB and 3D feature vectors into a shared low-dimensional space before using them as retrieval keys. We define two memory banks as

$$\begin{aligned} \mathcal{B}_{2D} &= \{\phi_{2D}(F_{2D}) \mid F_{2D} \in E_{2D}\}, \\ \mathcal{B}_{cross} &= \{(\phi_{3D}(F_{3D}^{(i)}), F_{2D}^{(i)})\}_{i=1}^{N}. \end{aligned} \quad (4)$$

$i$ denotes the $i$-th pair of $F_{3D}$ and $F_{2D}$ in the feature maps, with a total of $N$ nominal feature pairs. The $\mathcal{B}_{2D}$ is prepared for 2D feature filtering, and the $\mathcal{B}_{cross}$ is for crossmodal retrieval in the final stage. In order to construct a more compact representation of the clusters and reduce the size of memory banks, we use a minimax facility location coreset selection as (Sener & Savarese, 2018; Sinha et al., 2020; Roth et al., 2022) to approximately maintain the coverage of cluster anchors in feature space. We use the optimization formula below and utilize the same iterative greedy approximation in prior works (Sener & Savarese, 2018; Roth et al., 2022) for computation due to its NP-hard nature:

$$\mathcal{B}_{final}^* = \operatorname*{arg\,min}_{\mathcal{B}^* \subset \mathcal{B}} \max_{m \in \mathcal{B}} \min_{n \in \mathcal{B}^*} \|m - n\|_2. \quad (5)$$

Notably, only features considered as keys participate in clustering, and we use $m$ and $n$ as notations for keys in a specific memory bank. However, if a key is abandoned in the process, its corresponding feature value will also be dropped. By clustering features of the two modalities separately, we are able to obtain compact yet independent representations of 2D and 3D features. Specifically for $\mathcal{B}_{cross}^*$, more representative 3D features are available for search, while the 2D feature corresponding to it can have some redundancies, which serve the purpose of retrieving multiple candidates in one-to-many mapping situations.

**Attention-Aided Retrieval.** We use cosine similarity in the key space $\mathcal{B}_{cross}^*$ and perform nearest-neighbor matching to return the paired value. Given a 3D query $F_{3D}$, we retrieve the index

$$i^* = \arg\max_i \frac{\phi_{3D}(F_{3D})^\top f_{3D}^{(i)}}{\|\phi_{3D}(F_{3D})\| \|f_{3D}^{(i)}\|}, f_{3D}^{(i)} \in \mathcal{B}_{cross}^* \quad (6)$$

Then we use the index to acquire the $F_{2D}$ corresponding to this queried $F_{3D}$.

$$F_{2D}^* = F_{2D}^{(i^*)}, \quad (f_{3D}^{(i^*)}, F_{2D}^{(i^*)}) \in \mathcal{B}_{cross}^*. \quad (7)$$

Nearest-neighbor–based crossmodal retrieval is known to be brittle under ambiguous mappings, where a single 3D feature may correspond to multiple valid 2D appearances. Hard assignment to a single neighbor can therefore introduce high variance and propagate retrieval errors. To make this retrieval mechanism more dynamic and open to generalization, we retain the top-$K$ best 2D candidates and employ a lightweight scaled dot-product attention to adaptively aggregate them. For a ground truth nominal pair $(F_{3D}, F_{2D})$, we first apply the fixed linear projection to $F_{3D}$ to obtain $f_{3D}$ whose dimensionality matches the projected feature keys stored in $\mathcal{B}_{cross}^*$. We then use $f_{3D}$ as a query to retrieve the top-$K$ crossmodal feature pairs: $\{f_{3D}^{(i)}\}_{i=1}^{K}, \{F_{2D}^{(i)}\}_{i=1}^{K}$. Then we use a set of linear projections

$$\mathbf{q} = W_q f_{3D}, \quad \mathbf{k}_i = W_k F_{2D}^{(i)}, \quad \mathbf{v}_i = W_v F_{2D}^{(i)} \quad (8)$$

Then we obtain the attention weights using a scaled softmax operation. Here, $d$ denotes the dimensionality of the projected query and key embeddings used for attention.

$$\alpha_i = \frac{\exp\left(\frac{\mathbf{q}^\top \mathbf{k}_i}{\sqrt{d}}\right)}{\sum_{j=1}^{K} \exp\left(\frac{\mathbf{q}^\top \mathbf{k}_j}{\sqrt{d}}\right)}, \quad i = 1, \dots, K \quad (9)$$

Finally, by weighting retrieved 2D features using scaled dot-product attention, our approach produces a confidence-weighted reconstruction.

$$\hat{F}_{2D} = W_o \sum_{i=1}^{K} \alpha_i \mathbf{v}_i \quad (10)$$

The top-$K$ retrieval method avoids computing attention across the entire memory bank, which results in prohibitively high time and memory costs. This attention module is trained by optimizing the cosine distance between retrieved and ground truth features.

$$\mathcal{L}_r = \sum \left(1 - \frac{F_{2D} \cdot \hat{F}_{2D}}{\|F_{2D}\| \|\hat{F}_{2D}\|}\right). \quad (11)$$

### 3.3. Feature Replacement

This stage takes place at inference time, where we have trained the crossmodal mapping networks and retrieval modules. With $E_{3D}$ and $E_{2D}$ as feature maps extracted, we process $F_{3D}$ and $F_{2D}$ in pairs using (1) to obtain reconstructed feature maps $\hat{E}_{2D}^1$ and $\hat{E}_{3D}^1$. Since the one-to-many crossmodal ambiguity is primarily observed in the 3D-to-2D mapping direction, we only apply the subsequent filtering

and replacement operations to reconstructed 2D features. The $\hat{E}_{2D}^1$ is then sent into a prototype distance filter to identify potentially unreliable features using cosine distance with clusters in $\mathcal{B}_{2D}^*$. Intuitively, a reconstructed feature is considered unreliable if it lies far from all high-confidence 2D prototypes in the memory bank; thus, the filter is defined as follows. For any reconstructed $\hat{F}_{2D}$, we compute its cosine distance to the closest cluster center.

$$d_{\cos}(\hat{F}_{2D}, \mathcal{B}_{2D}^*) = \min_{b_j \in \mathcal{B}_{2D}^*} \left( 1 - \frac{\phi_{2D}(\hat{F}_{2D})^\top b_j}{\|\phi_{2D}(\hat{F}_{2D})\| \, \|b_j\|} \right). \quad (12)$$

Based on this distance, we define the discriminant function as follows

$$\mathcal{D}(\hat{F}_{2D}) = \begin{cases} 1, & d_{\cos}(\hat{F}_{2D}, \mathcal{B}_{2D}^*) > \theta, \\ 0, & \text{otherwise.} \end{cases} \quad (13)$$

where $\theta$ is a preset threshold and $\mathcal{D}(\hat{F}_{2D}) = 1$ indicates that the feature is unreliable and should be replaced. The $F_{3D}$ related to it will be used for Attention-Aided Retrieval and to obtain a new $\hat{F}_{2D}$ to replace it.

### 3.4. Aggregation

With the reconstructed feature maps $\hat{E}_{2D}$ and $\hat{E}_{3D}$, we compare them with the original feature maps using Euclidean distance $\mathcal{ED}$ as the pixel-wise discrepancy metric, following (Costanzino et al., 2024), to obtain anomaly maps.

$$\mathcal{A}_{2D} = \mathcal{ED}(\hat{E}_{2D}, E_{2D}), \mathcal{A}_{3D} = \mathcal{ED}(\hat{E}_{3D}, E_{3D}) \quad (14)$$

A high score in the anomaly map $\mathcal{A}$ suggests that the other modality disagrees with the feature of this pixel. We use a dot product to suppress modality-specific responses and emphasize regions where both modalities consistently indicate abnormality.

$$\mathcal{A} = \mathcal{A}_{3D} \odot \mathcal{A}_{2D} \quad (15)$$

Although asymmetric responses (e.g., low-high activation pairs) are attenuated by the dot product, anomaly inference is ultimately based on relative contrast rather than absolute magnitude. As a result, these regions can still remain distinguishable from normal low-low regions while reducing false positives caused by unilateral modality noise. The aggregated map is then filtered by a Gaussian kernel to smooth the pixel-wise scores. For sample-level anomaly detection results, the anomaly score is represented by the maximum score in the anomaly map $\mathcal{A}$.

## 4. Experiments

### 4.1. Experiment Setup

**Benchmarks.** We evaluate our method on two public benchmarks, MVTec 3D-AD (Bergmann et al., 2022) and Eyecandies (Bonfiglioli et al., 2022), following a few-shot normal-only protocol. For each class, only $K \in \{1, 2, 4\}$ normal

samples are used for training with a fixed random seed, and evaluation is conducted on the full test set. MVTec 3D-AD contains 10 real-world industrial object categories with paired RGB images and point clouds, while Eyecandies comprises 10 photo-realistic synthetic candy categories. Both datasets provide pixel-aligned RGB and 3D coordinates, enabling dense crossmodal correspondence.

**Metrics.** We adopt standard anomaly detection metrics. Image-level performance is measured by the Area Under the Receiver Operating Characteristic curve (I-AUROC). Pixel-level anomaly segmentation is evaluated using the Area Under the Per-Region Overlap (AUPRO) at a 30% false positive rate, following common practice.

**Implementation Details.** We employ the same feature extractors as (Wang et al., 2023b), which are DINO ViT-B/8 (Caron et al., 2021) trained on ImageNet (Deng et al., 2009) and Point-MAE (Pang et al., 2023) trained on ShapeNet (Chang et al., 2015). We resize the input RGB image to $224 \times 224$ to match DINO and normalize it using ImageNet statistics. The deep layer feature of shape $28 \times 28 \times 768$ is bilinearly interpolated to the image size $224 \times 224 \times 768$. 3D point clouds are encoded using Point-MAE into scattered point features, which are then interpolated following (Costanzino et al., 2024) into dense aligned feature maps of shape $224 \times 224 \times 1152$ before being fed to CFR.

$M_{2D \to 3D}$ and $M_{3D \to 2D}$ are implemented as lightweight GeGLUs, where the hidden dimension is set to the average of the 2D and 3D feature sizes. In Attention-Aided Retrieval, the key-space dimension is set to $d = 128$, and query, key, and value embeddings are projected to this shared space for efficient attention computation, then projected back to the original feature dimension. For coreset selection, we retain 10% of training features. Compared to prior baselines, CFR adopts a lightweight design that incurs only marginal computational and memory overhead. The threshold $\theta$ is set to a fixed value (0.4) based on similarity statistics observed on bagel class in MVTec 3D-AD, and is kept constant across all categories and datasets. We observe that CFR remains stable within a reasonable range of $\theta$ ($\pm 0.1$), indicating that the performance is not sensitive to this hyperparameter. A more detailed analysis of this threshold is provided in Section 4.4. For top-$K$ retrieval, we use $K = 3$. The networks are all trained for 100 epochs using the Adam optimizer with a learning rate initialized as $1 \times 10^{-3}$.

We conduct experiments using both our implementation and open-source code from prior anomaly detection works. All experiments are performed on a single NVIDIA GeForce RTX 4090D GPU.

*Table 1.* I-AUROC/30% AUPRO scores for anomaly detection and localization of all categories of MVTec 3D-AD. **Bold** indicates the best performance.

| Setting | Method | MVTec 3D-AD | | | | | | | | | | |
|---|---|---|---|---|---|---|---|---|---|---|---|---|
| | | Bagel | Cable Gland | Carrot | Cookie | Dowel | Foam | Peach | Potato | Rope | Tire | Mean |
| **1-shot** | AST (Rudolph et al., 2023) | 70.7/75.9 | 42.2/73.3 | 54.8/88.0 | 49.0/60.2 | 53.8/79.4 | 46.4/44.0 | 51.9/84.0 | 49.7/85.9 | 72.0/75.8 | 41.9/74.0 | 53.2/74.0 |
| | EasyNet (Chen et al., 2023) | 61.4/79.6 | 21.2/75.1 | 52.0/91.0 | 75.9/69.8 | 56.5/85.8 | 62.8/49.4 | 65.7/69.0 | 63.0/88.1 | **94.6**/71.8 | 47.7/75.4 | 60.1/75.5 |
| | ShapeGuided (Chu et al., 2023) | 65.9/95.9 | 44.4/71.6 | 62.3/93.5 | 93.8/**94.1** | 59.3/**86.4** | 57.6/63.8 | 67.6/94.0 | 42.8/96.3 | 93.3/88.8 | **62.9**/90.1 | 65.0/87.4 |
| | M3DM (Wang et al., 2023b) | 87.8/95.3 | 64.1/81.5 | **78.0/97.2** | 92.7/90.3 | **64.2**/81.6 | 65.3/82.0 | **75.5**/94.0 | **79.8**/94.8 | 85.8/95.2 | 45.4/89.8 | 73.9/90.2 |
| | CFM (Costanzino et al., 2024) | 43.5/92.6 | 56.7/79.4 | 76.4/96.9 | **95.4**/92.8 | 53.2/85.2 | 71.4/**87.8** | 63.0/94.9 | 64.2/96.1 | 91.8/**96.6** | 59.2/**91.4** | 67.5/91.4 |
| | CFR (Ours) | **96.7/97.2** | **66.9/86.8** | 67.1/96.7 | 94.0/91.6 | 63.0/84.1 | **71.7**/87.4 | 72.6/**96.8** | 60.4/**96.4** | 89.9/94.9 | 57.3/91.1 | **74.0/92.3** |
| **2-shot** | AST (Rudolph et al., 2023) | 71.9/75.9 | 43.4/74.0 | 54.5/87.8 | 50.8/62.2 | 53.7/79.5 | 46.1/43.6 | 51.6/83.7 | 50.4/85.6 | 75.8/76.5 | 40.2/72.8 | 53.8/74.2 |
| | EasyNet (Chen et al., 2023) | 47.6/77.8 | **76.1**/62.9 | 52.6/92.6 | 60.2/59.1 | 31.7/58.8 | 52.3/57.2 | 71.9/21.1 | **76.1**/15.2 | 61.2/43.1 | 51.2/4.7 | 58.1/49.3 |
| | ShapeGuided (Chu et al., 2023) | 47.9/96.6 | 46.0/73.2 | 60.5/96.5 | 95.9/**95.5** | 55.3/86.5 | 50.2/71.4 | 69.7/95.3 | 41.3/96.3 | **93.6**/89.3 | **79.3**/91.3 | 64.0/89.2 |
| | M3DM (Wang et al., 2023b) | 91.8/95.5 | 57.0/82.9 | 79.8/97.2 | 94.5/88.0 | 61.4/**87.0** | **79.5**/79.6 | **79.2**/95.1 | 75.1/94.2 | 92.8/95.5 | 54.1/91.0 | 76.5/90.6 |
| | CFM (Costanzino et al., 2024) | 82.6/95.3 | 65.0/80.3 | **80.3/97.9** | **97.4**/93.1 | 53.3/85.6 | 68.0/89.1 | 70.4/96.1 | 74.7/96.3 | 92.1/**96.9** | 64.0/**94.1** | 74.8/92.5 |
| | CFR (Ours) | **94.1/97.3** | 67.5/**85.8** | 74.1/97.0 | 97.1/92.5 | **65.7**/84.3 | 74.5/**92.6** | 70.8/**96.8** | 71.3/**97.4** | 91.8/95.3 | 64.4/92.0 | **77.1/93.1** |
| **4-shot** | AST (Rudolph et al., 2023) | 70.1/74.7 | 42.9/73.7 | 55.7/87.7 | 51.8/61.3 | 54.0/79.6 | 46.6/41.3 | 52.0/84.3 | 49.8/85.9 | 72.6/75.9 | 39.8/74.2 | 53.5/73.9 |
| | EasyNet (Chen et al., 2023) | 67.5/70.7 | 36.3/13.8 | 54.7/86.9 | 69.1/72.0 | 72.4/39.3 | 50.2/53.6 | 74.3/86.2 | 63.1/90.2 | 44.9/15.5 | 53.7/66.3 | 58.6/56.5 |
| | ShapeGuided (Chu et al., 2023) | 65.4/97.3 | 48.8/78.9 | 73.1/97.3 | 96.5/**95.4** | 69.8/**90.4** | 59.1/83.6 | 68.1/95.7 | 49.9/97.5 | 92.2/89.6 | **75.1**/92.1 | 69.8/91.8 |
| | M3DM (Wang et al., 2023b) | **98.6**/96.1 | 68.5/87.2 | 83.7/97.3 | 93.8/90.5 | 59.6/86.5 | 86.9/95.5 | 85.2/96.8 | 75.3/96.3 | 89.3/95.5 | 56.1/93.1 | 79.3/92.4 |
| | CFM (Costanzino et al., 2024) | 92.8/96.5 | 64.3/84.4 | **87.8/98.0** | **98.4**/93.5 | 64.1/89.7 | 77.8/**92.3** | 85.2/97.2 | 75.3/96.8 | **93.3/97.1** | 61.5/**94.8** | 80.1/94.0 |
| | CFR (Ours) | 97.6/**97.7** | **71.0/88.1** | 79.3/98.0 | 95.1/92.8 | **77.0**/89.0 | 76.1/92.1 | 83.4/97.1 | **78.1/97.7** | 91.8/95.4 | 55.1/93.9 | **80.5/94.2** |

*Table 2.* I-AUROC/30% AUPRO scores for anomaly detection and localization of all categories of Eyecandies. **Bold** indicates the best performance.

| Setting | Method | Eyecandies | | | | | | | | | | |
|---|---|---|---|---|---|---|---|---|---|---|---|---|
| | | C.Cane | C.Cookie | C.Praline | Confetto | Gummy.B | H.Truffle | Licorice.S | Lollipop | Marshm. | Peppermint.C | Mean |
| **1-shot** | M3DM (Wang et al., 2023b) | 36.2/86.8 | 66.9/82.5 | 73.1/70.6 | 84.8/94.2 | **71.3**/74.9 | 50.2/54.8 | 57.4/71.0 | **59.9**/84.2 | 60.3/89.8 | 81.0/**90.0** | 64.1/79.9 |
| | CFM (Costanzino et al., 2024) | 40.8/**91.4** | 48.8/81.5 | 73.1/69.7 | 88.8/91.9 | 55.1/**77.4** | 72.0/**66.1** | 45.4/63.6 | 59.4/82.9 | 76.5/85.1 | 57.4/80.9 | 61.7/79.1 |
| | CFR (Ours) | **47.4**/90.8 | **80.5/84.9** | **86.7/76.8** | **93.0/96.6** | 65.2/77.4 | **83.5**/55.5 | **82.9/79.4** | 42.6/**86.0** | **94.9/90.8** | **81.9**/89.2 | **75.9/82.7** |
| **2-shot** | M3DM (Wang et al., 2023b) | 38.9/81.1 | 67.5/84.4 | 81.1/68.6 | 92.3/**97.3** | 61.7/76.1 | 53.6/57.6 | 59.4/72.8 | 64.0/85.6 | 76.5/89.3 | **86.9**/90.8 | 68.2/80.4 |
| | CFM (Costanzino et al., 2024) | 38.2/**92.4** | 63.2/84.3 | 76.5/73.2 | 87.4/92.5 | 61.1/79.5 | **72.5**/70.3 | 51.8/64.9 | **64.9**/83.2 | 79.4/86.3 | 67.8/81.5 | 66.3/80.8 |
| | CFR (Ours) | **40.2**/91.3 | **86.2/86.0** | **87.8/76.2** | 92.8/96.3 | **71.5/81.5** | 72.0/59.3 | **85.3/84.7** | 49.3/**87.5** | **93.6/93.3** | 75.8/89.2 | **75.8/84.5** |
| **4-shot** | M3DM (Wang et al. 2023) | 42.4/82.4 | 74.9/84.9 | 78.7/72.1 | **91.4/96.8** | 70.2/80.2 | 53.4/61.7 | 80.2/**81.6** | **67.8/88.6** | 86.6/94.6 | **90.6/92.1** | 73.6/83.5 |
| | CFM (Costanzino et al., 2024) | 43.4/**93.0** | 71.4/85.1 | 78.1/75.0 | 85.8/94.1 | 72.3/81.8 | **73.0**/75.3 | 50.1/75.1 | 66.9/84.3 | **95.4**/93.6 | 79.2/85.2 | 71.6/84.3 |
| | CFR (Ours) | **54.9**/90.4 | **78.9/88.0** | **85.1/77.7** | 91.4/95.4 | **75.2/82.8** | 71.5/62.6 | **85.4**/81.5 | 65.1/86.4 | 90.9/94.0 | 81.0/88.2 | **77.9/84.7** |

## 4.2. Primary Results

**Few-shot Performance on MVTec 3D-AD.** Table 1 shows that CFR achieves the best results on mean I-AUROC and 30% AUPRO, which are image-level and pixel-level evaluation metrics, respectively, in all few-shot settings $\{1, 2, 4\}$-shot. For example, in the extremely challenging 1-shot setting, our method achieves an I-AUROC of 74.0% and 30% AUPRO of 92.3%. With three more training samples, i.e., in the 4-shot setting, I-AUROC increases by 6.5% and 30% AUPRO increases by 1.9%. These gains under strictly limited data settings highlight the effectiveness of CFR in industrial scenarios where samples are scarce or subject to confidentiality constraints.

**Few-shot Performance on Eyecandies.** Table 2 shows that CFR exceeds all competitors by a large margin on both metrics in all few-shot settings. In the 1-shot setting, our method achieves I-AUROC of 75.9% and 30% AUPRO of 82.7%, outperforming the second-best method by 11.8% and 2.8%, respectively. Notably, on Chocolate Praline, where chocolates share similar shapes but differ in frosting colors, and on Licorice Sandwich, which exhibits large color variation across layered yet visually similar semi-block structures, our model achieves substantial improvements. For Chocolate Praline and Licorice Sandwich, CFR improves I-AUROC by 13.6% and 25.5%, and AUPRO by 6.2% and 8.4%, respectively. These gains suggest that *our approach effectively*

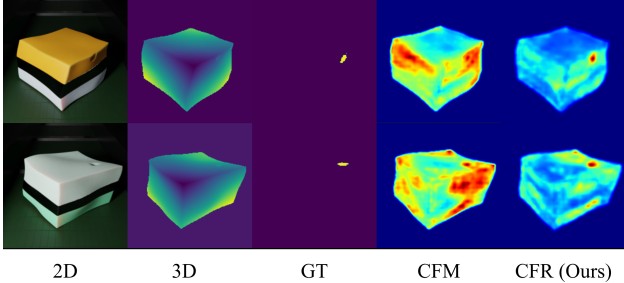

2D  3D  GT  CFM  CFR (Ours)

*Figure 3.* Qualitative examples for the *Licorice Sandwich*. GT denotes the ground truth. Colors ranging from blue to red indicate increasing anomaly scores at each location.

*resolves the one-to-many crossmodal mapping ambiguity*, demonstrating the effectiveness of the proposed method.

**Qualitative Analysis.** As shown in Figure 3, prior reconstruction-based methods suffer from severe false positives due to the one-to-many crossmodal mapping during training: a single 3D feature is forced to reconstruct multiple, distant 2D feature clusters with different colors, causing the network to converge to an averaged representation that matches none and triggers spurious anomaly responses. On *Licorice Sandwich*, this manifests as incorrectly marking large side regions as anomalous while preserving the top surface as normal, indicating overfitting to smooth geometry

*Table 3.* Low-FPR localization and image-level detection performance under few-shot settings of CFR / reconstruction baseline.

| DATASET | SHOT | 10% AUPRO | 5% AUPRO | 1% AUPRO | P-AUROC |
|---|---|---|---|---|---|
| EYECANDIES | 1-SHOT | **65.0** / 61.5 | **52.1** / 47.0 | **23.1** / 18.7 | **95.7** / 94.7 |
| | 2-SHOT | **68.1** / 66.8 | **55.8** / 53.2 | **25.6** / 23.3 | **96.3** / 95.6 |
| | 4-SHOT | **68.2** / 66.6 | **56.1** / 54.2 | **25.8** / 24.5 | **96.4** / 95.9 |
| MVTEC 3D-AD | 1-SHOT | **81.7** / 78.2 | **70.4** / 66.3 | **31.9** / 29.5 | **97.4** / 96.8 |
| | 2-SHOT | **82.8** / 80.2 | **72.3** / 69.6 | **33.9** / 32.6 | **97.5** / 97.0 |
| | 4-SHOT | **83.8** / 83.3 | **74.3** / 73.8 | **36.1** / 35.9 | **97.6** / 97.5 |

*Table 4.* AUROC / 30% AUPRO comparison in 1-shot between CFR with and without retrieval module in typical Eyecandies and MVTec 3D-AD categories exhibiting one-to-many correspondences. The best performance is highlighted in **bold**.

| CLASS | I-AUROC | | 30% AUPRO | |
|---|---|---|---|---|
| | +RET. | -RET. | +RET. | -RET. |
| C.COOKIE | **80.5** | 78.1 | **84.9** | 84.8 |
| C.PRALINE | **86.7** | 78.1 | **76.8** | 69.0 |
| CONFETTO | **93.0** | 89.9 | **96.6** | 94.8 |
| GUMMY BEAR | **65.2** | 57.5 | **77.4** | 73.3 |
| H.TRUFFLE | **83.5** | 69.4 | **55.5** | 55.3 |
| LICORICE.S | **82.9** | 72.5 | **79.4** | 73.7 |
| MARSHM. | **94.9** | 87.5 | **90.8** | 91.2 |
| PEPPERMINT.C | **81.9** | 80.6 | **89.2** | 84.8 |
| BAGEL | **96.7** | 92.7 | **97.2** | 96.4 |
| CABLE GLAND | **66.9** | 63.4 | **86.8** | 82.2 |
| DOWEL | **63.0** | 53.9 | **84.1** | 79.8 |
| PEACH | **72.6** | 66.4 | **96.8** | 96.2 |
| ROPE | **89.9** | 85.6 | **94.9** | 93.0 |
| MEAN | **81.3** | 75.1 | **85.4** | 82.6 |

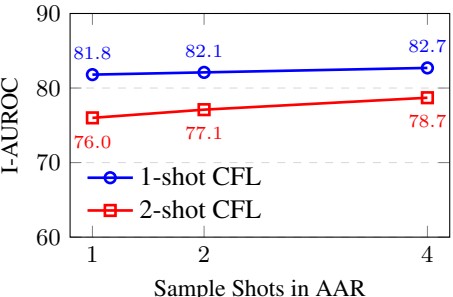

*Figure 4.* Crossmodal Feature Learning (CFL) combined with increasing numbers of Attention-Aided Retrieval (AAR) shots on the *Hazelnut Truffle* class, revealing a counterintuitive degradation in image-level detection performance as more training shots for crossmodal mappings are introduced.

with a monotone appearance and intolerance to valid color variations. In contrast, CFR explicitly detects such ambiguous reconstructions and replaces them with high-confidence features retrieved from training-time clusters, substantially reducing false positives.

### 4.3. Ablation Studies

**Replacement Analysis.** Table 4 shows that the Attention-Aided Retrieval module consistently improves both anomaly identification and localization across typical categories, yielding an average gain of 6.2% in I-AUROC and 2.8% in 30% AUPRO. For categories exhibiting pronounced one-to-many crossmodal ambiguity, such as *Chocolate Praline*, *Licorice Sandwich*, and *Marshmallow*, the Attention-Aided Retrieval module results in an average increase of 8.8% in I-AUROC. This ablation study indicates that the proposed coarse-to-fine framework effectively mitigates the one-to-many mapping issue by replacing ambiguous reconstructed features with high-confidence retrieved alternatives.

**Cyclic Reconstruction Analysis.** As shown in Table 4, CFR without feature retrieval and replacement still out-

performs its closest predecessor, CFM (Costanzino et al., 2024). This demonstrates the effectiveness of the proposed cyclic feature mapping, which regularizes crossmodal reconstruction functions by encouraging cycle-consistent feature correspondence. Notably, this design introduces no additional neural network modules and is implemented as a *plug-in* training loss, making it readily applicable to existing reconstruction-based anomaly detection methods.

**Effect of Increasing Training Shots on Hazelnut Truffle** An intriguing phenomenon is observed for the *Hazelnut Truffle* category in Table 2. While pixel-level segmentation performance consistently improves with more shots, the image-level anomaly identification performance (I-AUROC) unexpectedly degrades as the number of shots increases. Hazelnut Truffle objects (Figure 8) are roughly spherical with similar 3D geometry, while variations are dominated by surface color and gloss rather than shape. We attribute this counterintuitive behavior to an exacerbated one-to-many crossmodal mapping scenario: as more shots are introduced, a single 3D feature increasingly corresponds to multiple plausible RGB appearances, which reduces the discriminability of the learned mapping at the image level. This hypothesis is further supported by Figure 4. When we increase the number of samples used *only* in Attention-Aided Retrieval, while keeping the number of shots for Crossmodal Feature Learning fixed, image-level performance consistently improves. This suggests that the degradation

observed with more mapping shots stems from the ambiguity inherent in one-to-many crossmodal mappings rather than from data scarcity, and it highlights the effectiveness of our framework in mitigating this issue through selective feature replacement. More broadly, this observation serves as a cautionary note for future crossmodal mapping approaches: such methods can be particularly sensitive to *modality imbalance*, where naively increasing training samples may inadvertently amplify mapping ambiguity and harm anomaly identification performance.

### 4.4. Replacement Threshold Selection and Sensitivity

A key component of CFR is the Feature Replacement mechanism, which relies on a replacement threshold $\theta$. This threshold is applied to the cosine distance between a reconstructed feature and its nearest prototype in the clustered memory bank. Intuitively, if a reconstructed feature is insufficiently close to any reliable prototype, it is regarded as an uncertain reconstruction and replaced by the retrieved candidate feature.

In all experiments, $\theta$ is selected only once using a held-out normal category and then fixed for every dataset, class, and shot setting. We adopt this protocol because $\theta$ serves as a global control over replacement behavior rather than a category-specific hyperparameter. Unlike thresholds tied to raw feature magnitude or class statistics, cosine distance is inherently normalized, making $\theta$ transferable across different categories.

The threshold primarily reflects the reliability of the learned crossmodal mapping module: when reconstruction quality is high, reconstructed features remain close to memory-bank prototypes and replacement is rarely triggered; when reconstruction becomes ambiguous, larger cosine distance activates replacement. Therefore, $\theta$ is more related to the general mapping behavior of the network than to semantic properties of a specific class.

**Sensitivity Analysis.** We evaluate $\theta$ over the range $[0.0, 0.5]$ with a step size of $0.1$ on two randomly selected categories from each dataset (Eyecandies and MVTec 3D-AD) under the 4-shot setting. Table 5 reports the results. Across all tested categories, performance remains stable over a broad range of thresholds, with the best results typically achieved in $\theta \in [0.2, 0.4]$. The observed variation is minor, indicating that CFR is not sensitive to precise threshold tuning.

**Discussion.** The plateau behavior suggests that $\theta$ mainly controls whether obviously unreliable reconstructions are corrected, while the reconstructed feature maps remain largely stable under moderate changes within a reasonable range. In low-ambiguity categories, performance is nearly

*Table 5.* Per-class comparison across different threshold $\theta$ values. Different classes are grouped together. For each metric row, the best result across $\theta$ is highlighted in **bold**.

| Metric | $\theta = 0.0$ | $\theta = 0.1$ | $\theta = 0.2$ | $\theta = 0.3$ | $\theta = 0.4$ | $\theta = 0.5$ |
|---|---|---|---|---|---|---|
| **LicoriceSandwich** | | | | | | |
| I-AUROC | 0.773 | 0.789 | **0.854** | 0.845 | 0.821 | 0.821 |
| AUPRO@30% | 0.779 | 0.797 | **0.815** | 0.811 | 0.809 | 0.809 |
| AUPRO@10% | 0.614 | 0.607 | **0.634** | 0.633 | 0.621 | 0.621 |
| AUPRO@5% | 0.527 | 0.509 | **0.543** | 0.535 | 0.522 | 0.522 |
| AUPRO@1% | 0.246 | 0.226 | **0.249** | 0.245 | 0.234 | 0.234 |
| P-AUROC | 0.951 | 0.950 | 0.958 | 0.957 | **0.959** | **0.959** |
| **Marshmallow** | | | | | | |
| I-AUROC | 0.901 | 0.912 | 0.909 | **0.914** | 0.909 | 0.907 |
| AUPRO@30% | 0.926 | 0.932 | 0.938 | 0.932 | **0.943** | 0.932 |
| AUPRO@10% | 0.842 | 0.845 | **0.874** | 0.859 | 0.872 | 0.867 |
| AUPRO@5% | 0.762 | 0.766 | **0.808** | 0.788 | 0.802 | 0.800 |
| AUPRO@1% | 0.387 | 0.392 | **0.422** | 0.409 | 0.418 | 0.417 |
| P-AUROC | 0.984 | 0.988 | 0.992 | 0.993 | **0.994** | 0.993 |
| **Dowel** | | | | | | |
| I-AUROC | 0.757 | 0.759 | 0.764 | 0.762 | **0.771** | 0.754 |
| AUPRO@30% | 0.823 | 0.824 | 0.832 | 0.834 | **0.835** | **0.835** |
| AUPRO@10% | 0.668 | 0.671 | 0.698 | 0.704 | **0.706** | **0.706** |
| AUPRO@5% | 0.543 | 0.540 | 0.576 | 0.590 | **0.595** | **0.595** |
| AUPRO@1% | 0.221 | 0.217 | 0.242 | 0.247 | **0.256** | **0.256** |
| P-AUROC | 0.906 | 0.907 | 0.910 | **0.911** | **0.911** | **0.911** |
| **Peach** | | | | | | |
| I-AUROC | 0.720 | 0.764 | **0.839** | 0.838 | 0.838 | 0.838 |
| AUPRO@30% | 0.951 | 0.958 | **0.971** | 0.970 | 0.970 | 0.970 |
| AUPRO@10% | 0.854 | 0.874 | **0.914** | 0.911 | 0.910 | 0.910 |
| AUPRO@5% | 0.732 | 0.764 | **0.834** | 0.829 | 0.826 | 0.826 |
| AUPRO@1% | 0.305 | 0.338 | **0.405** | 0.399 | 0.396 | 0.396 |
| P-AUROC | 0.987 | 0.990 | **0.993** | **0.993** | **0.993** | **0.993** |

invariant to $\theta$; in categories with stronger one-to-many correspondence ambiguity, $\theta$ becomes moderately more influential, yet remains robust due to bounded replacement frequency.

Overall, these results demonstrate that CFR has low sensitivity to threshold selection, and a single globally selected $\theta$ generalizes well across datasets and categories without requiring per-class calibration.

## 5. Conclusion

In this work, we propose CFR, a multimodal anomaly detection framework that addresses the fundamental one-to-many mapping challenge. First, CFR introduces a plug-in cycle-consistent mapping loss to regularize crossmodal correspondence. Second, CFR adopts a novel coarse-to-fine reconstruction strategy that enforces feature fidelity and mitigates ambiguous crossmodal reconstructions. Third, CFR presents a crossmodal retrieval mechanism that avoids the hard assignment limitation of conventional memory bank methods. Finally, CFR effectively operates under data-scarce regimes and achieves state-of-the-art performance in few-shot anomaly detection.

## Acknowledgments

This work was supported in part by the National Natural Science Foundation of China (62306197); China Postdoctoral Science Foundation (2021TQ0223, 2022M712236); Postdoctoral Joint Training Program of Sichuan University (SCDXLHPY2307).

## Impact Statement

This paper presents work whose goal is to advance the field of Machine Learning. There are many potential societal consequences of our work, none of which we feel must be specifically highlighted here.

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

# A. Cyclic Feature Mapping Modules

We propose CFR as a flexible framework in which individual modules can be replaced with alternative implementations. In this section, we investigate different designs for the projection network used in the Cyclic Feature Mapping module, including (i) SwiGLU, (ii) GeGLU (the default architecture used in the main paper), (iii) MLP, (iv) ResMLP, (v) Token Transformer and (vi) Token FNet. In the following settings, all variants are controlled to have comparable model capacity, with approximately 2.6M-2.9M parameters for $M_{2D \to 3D}$ and $M_{3D \to 2D}$.

SwiGLU and GeGLU adopt a gated feed-forward architecture with different activation functions. Given an input feature, the network first projects it into gate and value representations. The hidden feature is then obtained by applying the activation function to the gate branch and performing element-wise multiplication with the value branch. Finally, the gated representation is passed through a linear layer to produce the output feature. For both SwiGLU and GeGLU, we use two mapping networks for the two crossmodal directions, with layer dimensions of [(768, 960 × 2), (960, 1152)] and [(1152, 960 × 2), (960, 768)], respectively. SwiGLU uses the SiLU activation function, whereas GeGLU uses GELU.

For the MLP architecture, we use two layers of nonlinear projections with GELU activations, followed by a final linear layer that maps the hidden representation to the required output dimension. For the two mapping directions, the layer dimensions are [(768, 960), (960, 960), (960, 1152)] and [(1152, 960), (960, 960), (960, 768)], respectively.

ResMLP is a residual multilayer perceptron variant that enhances feature transformation capacity through stacked residual fully connected blocks. The input feature is first projected into a hidden representation, followed by several nonlinear projection blocks. In each block, the projected feature is added to the block input through a residual connection, which helps stabilize optimization and preserve feature information across depth. The final hidden representation is then linearly projected to the target modality space. For the two crossmodal mapping directions, we use asymmetric configurations to match their different dimensionalities. The 2D-to-3D mapping uses a hidden dimension of 661 with 3 residual blocks, while the 3D-to-2D mapping uses a hidden dimension of 526 with 7 residual blocks. GELU is adopted as the activation function throughout the network.

For the Token Transformer architecture, we first project the input feature into a sequence of latent tokens, where each token is a low-dimensional sub-representation of the original feature. These tokens are then processed by a lightweight Transformer encoder to model interactions across different parts of the feature through self-attention. After Transformer encoding, the token sequence is aggregated by mean pooling, and the pooled representation is finally projected to the target modality dimension. For the two mapping directions, we use different tokenization settings to match the input dimensionality. In the 2D-to-3D direction, the input is divided into 4 tokens of dimension 192, followed by a Transformer with hidden dimension 1152, 3 encoder layers, and 1 attention head. In the 3D-to-2D direction, the input is divided into 6 tokens of dimension 192, followed by a Transformer with hidden dimension 1536, 2 encoder layers, and 1 attention head. GELU is used as the activation function in the Transformer feed-forward blocks.

For the TokenFNet architecture, we first project the input feature into a sequence of latent tokens, where each token represents a compact sub-feature of the original input. Instead of using self-attention, TokenFNet processes these tokens with an FNet-style encoder stack, which mixes information across tokens through Fourier-based transformations together with feed-forward layers. After token mixing, the token sequence is aggregated by mean pooling, and the pooled representation is finally projected to the target modality dimension. For the two mapping directions, we use different tokenization and encoder configurations. In the 2D-to-3D direction, the input is divided into 4 tokens of dimension 192, followed by an FNet encoder with hidden dimension 1536 and 3 layers. In the 3D-to-2D direction, the input is divided into 4 tokens of dimension 288, followed by an FNet encoder with hidden dimension 576 and 4 layers. This design replaces attention-based token interaction with lightweight Fourier token mixing while preserving the same tokenized mapping pipeline.

*Table 6.* Per-class comparison across architectures. Best values are highlighted in **bold**.

| Metric | CandyCane | ChocolateCookie | ChocolatePraline | Confetto | GummyBear | HazelnutTruffle | LicoriceSandwich | Lollipop | Marshmallow | PeppermintCandy | Mean |
|---|---|---|---|---|---|---|---|---|---|---|---|
| **SwiGLU** | | | | | | | | | | | |
| I-AUROC | 0.586 | 0.800 | **0.909** | 0.918 | **0.761** | 0.755 | 0.835 | **0.673** | 0.894 | 0.818 | **0.795** |
| AUPRO@30% | 0.899 | 0.875 | 0.772 | 0.950 | **0.831** | 0.610 | 0.806 | 0.867 | 0.937 | 0.877 | 0.842 |
| AUPRO@10% | 0.698 | 0.717 | 0.634 | 0.854 | **0.666** | 0.364 | 0.613 | 0.602 | 0.872 | **0.720** | 0.674 |
| AUPRO@5% | 0.418 | 0.598 | 0.550 | 0.757 | **0.572** | 0.285 | 0.519 | 0.409 | 0.803 | 0.616 | 0.553 |
| AUPRO@1% | 0.071 | 0.275 | 0.276 | 0.381 | **0.274** | 0.133 | 0.243 | 0.121 | 0.420 | 0.287 | 0.248 |
| P-AUROC | **0.969** | 0.975 | **0.940** | 0.988 | 0.925 | 0.929 | 0.956 | **0.971** | 0.992 | 0.974 | 0.962 |
| **GeGLU** | | | | | | | | | | | |
| I-AUROC | 0.549 | 0.789 | 0.851 | 0.914 | 0.752 | 0.715 | **0.854** | 0.651 | **0.909** | 0.810 | 0.779 |
| AUPRO@30% | **0.904** | 0.880 | 0.777 | 0.954 | 0.828 | **0.626** | 0.815 | 0.864 | **0.940** | 0.882 | **0.847** |
| AUPRO@10% | **0.711** | **0.729** | **0.654** | 0.866 | 0.654 | **0.392** | 0.634 | 0.591 | **0.879** | 0.713 | **0.682** |
| AUPRO@5% | **0.438** | 0.613 | **0.572** | 0.769 | 0.556 | **0.310** | 0.540 | 0.389 | **0.816** | 0.608 | **0.561** |
| AUPRO@1% | 0.077 | **0.298** | **0.293** | 0.374 | 0.273 | **0.147** | 0.240 | 0.111 | **0.433** | 0.286 | **0.253** |
| P-AUROC | **0.969** | 0.976 | 0.939 | 0.990 | 0.926 | **0.936** | 0.959 | 0.970 | **0.993** | **0.978** | **0.964** |
| **MLP** | | | | | | | | | | | |
| I-AUROC | **0.619** | 0.776 | 0.854 | 0.854 | 0.657 | 0.773 | 0.683 | 0.618 | 0.811 | 0.826 | 0.747 |
| AUPRO@30% | 0.892 | 0.876 | 0.755 | 0.939 | 0.817 | 0.594 | 0.786 | 0.867 | 0.938 | 0.847 | 0.831 |
| AUPRO@10% | 0.677 | 0.715 | 0.616 | 0.830 | 0.626 | 0.340 | 0.577 | 0.602 | 0.857 | 0.654 | 0.649 |
| AUPRO@5% | 0.379 | 0.594 | 0.531 | 0.724 | 0.526 | 0.259 | 0.462 | 0.392 | 0.777 | 0.556 | 0.520 |
| AUPRO@1% | 0.067 | 0.278 | 0.264 | 0.356 | 0.249 | 0.115 | 0.204 | 0.114 | 0.379 | 0.269 | 0.229 |
| P-AUROC | 0.967 | 0.975 | 0.937 | 0.983 | 0.924 | 0.920 | 0.948 | 0.965 | 0.990 | 0.958 | 0.957 |
| **ResMLP** | | | | | | | | | | | |
| I-AUROC | 0.570 | **0.805** | 0.891 | **0.963** | 0.753 | 0.726 | 0.747 | 0.631 | 0.826 | 0.842 | 0.775 |
| AUPRO@30% | 0.897 | **0.881** | 0.764 | 0.958 | 0.820 | 0.593 | 0.795 | **0.872** | 0.932 | 0.877 | 0.839 |
| AUPRO@10% | 0.691 | 0.723 | 0.623 | **0.881** | 0.647 | 0.338 | 0.603 | **0.616** | 0.866 | 0.719 | 0.671 |
| AUPRO@5% | 0.406 | **0.615** | 0.539 | **0.795** | 0.541 | 0.264 | 0.496 | 0.413 | 0.794 | 0.606 | 0.547 |
| AUPRO@1% | 0.086 | 0.297 | 0.268 | **0.404** | 0.246 | 0.124 | 0.214 | 0.135 | 0.405 | 0.275 | 0.245 |
| P-AUROC | **0.969** | **0.978** | **0.940** | 0.989 | 0.927 | **0.936** | 0.952 | 0.970 | 0.991 | 0.974 | 0.963 |
| **Token Transformer** | | | | | | | | | | | |
| I-AUROC | 0.610 | 0.774 | 0.859 | 0.946 | 0.731 | **0.789** | 0.683 | 0.636 | 0.792 | 0.856 | 0.768 |
| AUPRO@30% | 0.902 | 0.871 | **0.783** | 0.956 | 0.811 | 0.580 | **0.830** | 0.871 | 0.935 | 0.881 | 0.842 |
| AUPRO@10% | 0.706 | 0.698 | 0.632 | 0.873 | 0.625 | 0.357 | **0.649** | 0.613 | 0.864 | 0.719 | 0.674 |
| AUPRO@5% | 0.433 | 0.578 | 0.547 | 0.782 | 0.514 | 0.282 | **0.545** | 0.424 | 0.792 | 0.610 | 0.551 |
| AUPRO@1% | **0.096** | 0.271 | 0.275 | 0.390 | 0.236 | 0.127 | **0.244** | **0.149** | 0.399 | 0.285 | 0.247 |
| P-AUROC | **0.969** | 0.974 | 0.937 | **0.991** | 0.928 | 0.927 | **0.962** | 0.966 | 0.992 | 0.976 | 0.962 |
| **Token FNet** | | | | | | | | | | | |
| I-AUROC | 0.595 | 0.750 | 0.861 | 0.930 | 0.760 | 0.749 | 0.734 | 0.642 | 0.875 | **0.858** | 0.775 |
| AUPRO@30% | 0.901 | 0.871 | 0.775 | **0.959** | 0.807 | 0.558 | 0.816 | 0.865 | **0.940** | 0.878 | 0.837 |
| AUPRO@10% | 0.702 | 0.704 | 0.626 | 0.880 | 0.639 | 0.337 | 0.625 | 0.594 | 0.853 | **0.720** | 0.668 |
| AUPRO@5% | 0.430 | 0.587 | 0.541 | 0.784 | 0.536 | 0.261 | 0.522 | 0.415 | 0.780 | **0.617** | 0.547 |
| AUPRO@1% | **0.096** | 0.281 | 0.268 | 0.378 | 0.257 | 0.116 | **0.244** | 0.147 | 0.399 | **0.294** | 0.248 |
| P-AUROC | **0.969** | 0.976 | 0.935 | 0.989 | 0.921 | 0.917 | 0.959 | 0.964 | 0.990 | 0.975 | 0.959 |

# B. Inference Time and Memory

Computational efficiency is important for anomaly detection in industrial settings. We investigate the memory footprint and inference speed of CFR and the extra inference time and memory required by our Attention-Aided Retrieval module by providing profiling results of Candy Cane (Table 7) in the Eyecandies dataset to quantify the inference overhead of the proposed retrieval and replacement modules. All measurements are conducted on a single NVIDIA GeForce RTX 4090D GPU. Collecting such profiling statistics introduces additional runtime overhead, so the measured latency is slightly higher than normal inference.

We compare the full model with a reconstruction-only variant that removes retrieval and replacement while keeping the same backbones and feature mapping modules. The reconstruction-only variant runs at 16.6 FPS / 60.2 ms latency with 1.50 GB peak GPU memory. Our full model runs at 8.4 FPS / 118.5 ms with 2.97 GB peak memory under profiling, corresponding to +58.3 ms latency ($\sim$1.97$\times$) and +1.47 GB memory ($\sim$1.97$\times$). We note that profiling introduces synchronization overhead, and the practical inference speed is approximately 12 FPS.

The persistent memory overhead introduced by the proposed retrieval components remains limited. The feature extractor occupies 410.6 MB, while the additional modules are lightweight, including Cyclic Feature Mapping ($\sim$21 MB), memory bank (58.8 MB), KV bank ($\sim$147 MB), and attention retriever (1.9 MB).

*Table 7.* Detailed profiling results on CandyCane (4-shot). Latency is averaged per call. GPU memory is measured by peak allocated memory during each module execution.

| Module | Calls | Total Time (s) | Avg Latency (ms) | FPS | Avg GPU Peak (MB) | Max GPU Peak (MB) |
|---|---|---|---|---|---|---|
| anomaly_scoring | 40 | 0.001 | 0.03 | 33974.5 | 2966.7 | 2966.7 |
| attention_fusion | 11 | 0.002 | 0.16 | 6157.3 | 2966.7 | 2966.7 |
| cfm_2d_to_3d | 40 | 0.204 | 5.11 | 195.7 | 2966.7 | 2966.7 |
| cfm_3d_to_2d | 40 | 0.333 | 8.32 | 120.2 | 2966.7 | 2966.7 |
| data_loading | 40 | 0.020 | 0.49 | 2041.7 | 0.0 | 0.0 |
| end_to_end | 40 | 4.740 | 118.50 | 8.4 | 2966.7 | 2966.7 |
| feature_resize_align | 40 | 0.009 | 0.22 | 4485.0 | 2966.7 | 2966.7 |
| host_to_device | 40 | 0.011 | 0.28 | 3569.6 | 1706.9 | 1706.9 |
| kv_bank_sampling | 11 | 0.001 | 0.06 | 16418.2 | 2966.7 | 2966.7 |
| kv_bank_topk_retrieval | 11 | 0.003 | 0.27 | 3747.1 | 2966.7 | 2966.7 |
| memory_bank_distance | 40 | 2.372 | 59.29 | 16.9 | 2586.8 | 2586.8 |
| pc_nonzero_filter | 40 | 0.002 | 0.06 | 17821.6 | 2450.4 | 2450.4 |
| point_interpolation | 40 | 0.064 | 1.59 | 628.2 | 2450.4 | 2450.4 |
| rgb_backbone | 40 | 0.274 | 6.86 | 145.7 | 1706.9 | 1706.9 |
| 3D_backbone | 40 | 0.769 | 19.23 | 52.0 | 2450.4 | 2450.4 |

*Table 8.* Module-wise runtime and memory profiling.

| Module | Latency (ms) | Ratio (%) | Peak GPU (MB) |
|---|---|---|---|
| Memory-bank distance | 59.3 | 50.0 | 2586.8 |
| 3D backbone | 19.2 | 16.2 | 2450.4 |
| Feature Mapping | 13.4 | 11.3 | 2966.7 |
| RGB backbone | 6.9 | 5.8 | 1706.9 |
| Others | 19.7 | 16.7 | – |
| End-to-end | 118.5 | 100.0 | 2966.7 |

A representative memory-bank baseline requires approximately 6 GB GPU memory and runs at around 1 FPS. In comparison, our method uses $\sim$3.0 GB peak memory and achieves 8.4 FPS under profiling. The dominant cost comes from dense nearest-neighbor distance computation (59.3 ms, 50.0%), while memory-bank sampling, top-$K$ retrieval, and attention-based replacement each require less than 1 ms. The higher peak memory mainly comes from temporary buffers for dense distance computation rather than stored parameters or banks. The runtime breakdown is summarized in Table 8.

Overall, the proposed retrieval and replacement modules introduce moderate overhead over pure reconstruction, while remaining substantially more efficient than standard memory-bank approaches.

## C. Robustness Against Imperfect Alignment

Multimodal anomaly detection in real-world settings often suffers from imperfect crossmodal alignment. In our setting, this means that RGB pixels and point-cloud locations are not perfectly pairwise registered, but instead contain noise in their correspondence due to calibration bias or sensor mismatch.

To evaluate the robustness of CFR under such conditions, we replace the default pixel-aligned pairing with a fixed global RGB offset throughout the entire few-shot pipeline, simulating a persistent calibration error. Specifically, we apply a constant 2D shift $(d_x, d_y)$ to RGB features while keeping 3D coordinates unchanged as anchors. Overlap cropping is adopted instead of circular shifting, such that only truly overlapping regions are preserved and wrap-around artifacts are avoided. Importantly, the same offset is consistently applied during cyclic feature mapping training, memory-bank construction, attention retriever training, and inference, providing a more realistic setting with systematic misregistration.

Table 9 reports the class-averaged results on Eyecandies under different offsets, while Table 10 reports the full results. As expected, performance gradually decreases as the misalignment becomes larger, since dense crossmodal correspondence becomes less accurate. Nevertheless, CFR remains reasonably stable under moderate offsets. For example, under a shared offset of 4 px, the performance degradation is limited across all shot settings. Even with a larger offset of 8 px, the model still remains functional rather than collapsing entirely. These results suggest that CFR is robust to moderate systematic alignment noise, while severe or spatially varying misregistration remains a more challenging case for future work.

*Table 9.* Robustness against imperfect alignment on Eyecandies. Results are averaged over classes.

| Shot | Offset | AUPRO@30% | I-AUROC |
|---|---|---|---|
| 1-shot | 0 px | 0.827 | 0.759 |
| 1-shot | 4 px | 0.773 | 0.706 |
| 1-shot | 8 px | 0.684 | 0.696 |
| 2-shot | 0 px | 0.845 | 0.758 |
| 2-shot | 4 px | 0.804 | 0.728 |
| 2-shot | 8 px | 0.730 | 0.698 |
| 4-shot | 0 px | 0.847 | 0.779 |
| 4-shot | 4 px | 0.805 | 0.761 |
| 4-shot | 8 px | 0.730 | 0.699 |

*Table 10.* Per-class comparison across different $(s_x, s_y)$ settings.

| Shot | Metric | CandyCane | ChocolateCookie | ChocolatePraline | Confetto | GummyBear | HazelnutTruffle | LicoriceSandwich | Lollipop | Marshmallow | PeppermintCandy | Mean |
|---|---|---|---|---|---|---|---|---|---|---|---|---|
| | | | | | | $s_x = 4, s_y = 4$ | | | | | | |
| 1-shot | I-AUROC | 0.414 | 0.670 | 0.888 | 0.952 | 0.527 | 0.624 | 0.773 | 0.461 | 0.899 | 0.853 | 0.706 |
| | AUPRO@30% | 0.909 | 0.793 | 0.702 | 0.915 | 0.729 | 0.515 | 0.704 | 0.837 | 0.843 | 0.778 | 0.772 |
| | AUPRO@10% | 0.726 | 0.513 | 0.512 | 0.774 | 0.460 | 0.266 | 0.434 | 0.514 | 0.676 | 0.575 | 0.545 |
| | AUPRO@5% | 0.472 | 0.359 | 0.426 | 0.664 | 0.320 | 0.189 | 0.320 | 0.291 | 0.568 | 0.478 | 0.409 |
| | AUPRO@1% | 0.106 | 0.130 | 0.200 | 0.321 | 0.113 | 0.073 | 0.118 | 0.060 | 0.266 | 0.216 | 0.160 |
| | P-AUROC | 0.969 | 0.952 | 0.919 | 0.983 | 0.900 | 0.901 | 0.914 | 0.952 | 0.964 | 0.945 | 0.940 |
| 2-shot | I-AUROC | 0.368 | 0.802 | 0.834 | 0.928 | 0.663 | 0.627 | 0.802 | 0.517 | 0.902 | 0.838 | 0.728 |
| | AUPRO@30% | 0.914 | 0.813 | 0.713 | 0.907 | 0.768 | 0.576 | 0.769 | 0.866 | 0.900 | 0.818 | 0.804 |
| | AUPRO@10% | 0.741 | 0.554 | 0.537 | 0.768 | 0.568 | 0.294 | 0.505 | 0.600 | 0.787 | 0.626 | 0.598 |
| | AUPRO@5% | 0.502 | 0.406 | 0.442 | 0.644 | 0.455 | 0.204 | 0.374 | 0.379 | 0.695 | 0.535 | 0.464 |
| | AUPRO@1% | 0.139 | 0.154 | 0.198 | 0.302 | 0.195 | 0.081 | 0.155 | 0.105 | 0.338 | 0.255 | 0.192 |
| | P-AUROC | 0.969 | 0.959 | 0.922 | 0.981 | 0.926 | 0.912 | 0.942 | 0.955 | 0.982 | 0.957 | 0.951 |
| 4-shot | I-AUROC | 0.550 | 0.699 | 0.862 | 0.923 | 0.731 | 0.698 | 0.810 | 0.594 | 0.904 | 0.834 | 0.760 |
| | AUPRO@30% | 0.908 | 0.836 | 0.714 | 0.905 | 0.781 | 0.590 | 0.762 | 0.852 | 0.909 | 0.796 | 0.805 |
| | AUPRO@10% | 0.723 | 0.598 | 0.548 | 0.759 | 0.579 | 0.326 | 0.543 | 0.557 | 0.805 | 0.591 | 0.603 |
| | AUPRO@5% | 0.461 | 0.452 | 0.461 | 0.646 | 0.466 | 0.227 | 0.415 | 0.393 | 0.715 | 0.498 | 0.473 |
| | AUPRO@1% | 0.095 | 0.185 | 0.218 | 0.311 | 0.201 | 0.092 | 0.161 | 0.128 | 0.346 | 0.232 | 0.197 |
| | P-AUROC | 0.972 | 0.966 | 0.927 | 0.981 | 0.934 | 0.924 | 0.943 | 0.963 | 0.983 | 0.948 | 0.954 |
| | | | | | | | $s_x = 8, s_y = 8$ | | | | | | |
| 1-shot | I-AUROC | 0.523 | 0.693 | 0.734 | 0.931 | 0.566 | 0.594 | 0.710 | 0.483 | 0.885 | 0.840 | 0.696 |
| | AUPRO@30% | 0.907 | 0.680 | 0.611 | 0.852 | 0.660 | 0.466 | 0.572 | 0.822 | 0.655 | 0.612 | 0.684 |
| | AUPRO@10% | 0.722 | 0.297 | 0.374 | 0.650 | 0.363 | 0.198 | 0.259 | 0.469 | 0.411 | 0.351 | 0.409 |
| | AUPRO@5% | 0.472 | 0.163 | 0.274 | 0.540 | 0.238 | 0.112 | 0.160 | 0.237 | 0.307 | 0.250 | 0.275 |
| | AUPRO@1% | 0.124 | 0.036 | 0.106 | 0.251 | 0.081 | 0.030 | 0.047 | 0.030 | 0.127 | 0.094 | 0.093 |
| | P-AUROC | 0.966 | 0.913 | 0.885 | 0.966 | 0.882 | 0.868 | 0.870 | 0.945 | 0.907 | 0.903 | 0.911 |
| 2-shot | I-AUROC | 0.373 | 0.816 | 0.680 | 0.920 | 0.585 | 0.520 | 0.730 | 0.562 | 0.907 | 0.891 | 0.698 |
| | AUPRO@30% | 0.903 | 0.704 | 0.625 | 0.837 | 0.715 | 0.516 | 0.655 | 0.839 | 0.781 | 0.724 | 0.730 |
| | AUPRO@10% | 0.710 | 0.338 | 0.416 | 0.641 | 0.446 | 0.237 | 0.340 | 0.520 | 0.600 | 0.492 | 0.474 |
| | AUPRO@5% | 0.454 | 0.195 | 0.301 | 0.522 | 0.339 | 0.138 | 0.214 | 0.304 | 0.493 | 0.386 | 0.335 |
| | AUPRO@1% | 0.100 | 0.042 | 0.110 | 0.235 | 0.125 | 0.037 | 0.058 | 0.062 | 0.202 | 0.157 | 0.113 |
| | P-AUROC | 0.967 | 0.922 | 0.895 | 0.964 | 0.910 | 0.883 | 0.901 | 0.947 | 0.947 | 0.929 | 0.927 |
| 4-shot | I-AUROC | 0.531 | 0.579 | 0.800 | 0.846 | 0.628 | 0.638 | 0.725 | 0.496 | 0.906 | 0.842 | 0.699 |
| | AUPRO@30% | 0.901 | 0.735 | 0.629 | 0.843 | 0.726 | 0.501 | 0.628 | 0.846 | 0.784 | 0.704 | 0.730 |
| | AUPRO@10% | 0.702 | 0.374 | 0.433 | 0.652 | 0.460 | 0.218 | 0.336 | 0.538 | 0.614 | 0.461 | 0.479 |
| | AUPRO@5% | 0.431 | 0.195 | 0.344 | 0.535 | 0.340 | 0.135 | 0.205 | 0.351 | 0.514 | 0.368 | 0.342 |
| | AUPRO@1% | 0.069 | 0.043 | 0.153 | 0.251 | 0.139 | 0.038 | 0.054 | 0.090 | 0.226 | 0.145 | 0.121 |
| | P-AUROC | 0.969 | 0.930 | 0.901 | 0.969 | 0.922 | 0.898 | 0.900 | 0.960 | 0.945 | 0.920 | 0.931 |

# D. Single-Modality Ablation

To better understand the contribution of the crossmodal aggregation strategy in Section 3.4, we perform an ablation by disabling the final joint scoring step and using only a single modality branch for inference. Concretely, instead of combining the two discrepancy maps as

$$\mathcal{A} = \mathcal{A}_{3D} \odot \mathcal{A}_{2D}, \tag{16}$$

we alternatively use only the RGB branch anomaly map $\mathcal{A}_{2D}$ or only the 3D branch anomaly map $\mathcal{A}_{3D}$ for prediction. We conduct the study on MVTec 3D-AD under the 4-shot setting. Results are summarized in Table 11. The full crossmodal score consistently achieves the best performance in mean over all classes, obtaining 94.2 AUPRO@30% and 80.3 I-AUROC.

Using only RGB information causes a moderate drop to 92.2 / 78.2, while relying only on 3D information leads to a larger degradation of 91.4 / 69.8.

*Table 11.* Comparison of single-modality scores and full multimodal score on MVTec 3D-AD 4-shot setting. Best result within each class/metric block is highlighted in **bold**. Mean denotes the average over all classes.

| Metric | Type | Bagel | Cable_Gland | Carrot | Cookie | Dowel | Foam | Peach | Potato | Rope | Tire | Mean |
|---|---|---|---|---|---|---|---|---|---|---|---|---|
| AUPRO@30% | rgb_score | 0.969 | 0.881 | 0.977 | 0.879 | 0.819 | 0.865 | 0.971 | 0.971 | 0.935 | **0.949** | 0.922 |
| | 3D_score | 0.969 | 0.833 | 0.976 | 0.916 | 0.821 | 0.901 | 0.955 | 0.974 | 0.938 | 0.857 | 0.914 |
| | full_score | **0.977** | **0.881** | **0.980** | **0.928** | **0.890** | **0.921** | **0.971** | **0.977** | **0.954** | 0.939 | **0.942** |
| AUPRO@10% | rgb_score | 0.915 | **0.699** | 0.933 | 0.770 | 0.686 | 0.744 | **0.921** | 0.912 | 0.873 | **0.847** | 0.830 |
| | 3D_score | 0.911 | 0.610 | 0.929 | 0.850 | 0.680 | **0.796** | 0.862 | 0.921 | 0.885 | 0.736 | 0.818 |
| | full_score | **0.934** | 0.681 | **0.939** | **0.874** | **0.704** | **0.796** | 0.913 | **0.930** | **0.902** | 0.820 | **0.849** |
| AUPRO@1% | rgb_score | 0.423 | 0.199 | 0.439 | 0.288 | 0.228 | 0.294 | **0.412** | 0.406 | 0.385 | **0.340** | 0.341 |
| | 3D_score | 0.414 | 0.132 | 0.427 | 0.391 | 0.223 | 0.235 | 0.305 | 0.414 | 0.387 | 0.115 | 0.304 |
| | full_score | **0.462** | **0.209** | **0.459** | **0.436** | **0.243** | **0.321** | 0.398 | **0.442** | **0.428** | 0.305 | **0.370** |
| P-AUROC | rgb_score | 0.990 | 0.963 | 0.993 | 0.924 | 0.906 | 0.962 | 0.992 | 0.991 | 0.990 | **0.987** | 0.970 |
| | 3D_score | 0.990 | 0.950 | 0.993 | 0.936 | 0.906 | 0.975 | 0.987 | 0.993 | 0.990 | 0.957 | 0.968 |
| | full_score | **0.995** | **0.965** | **0.996** | **0.947** | **0.930** | **0.980** | **0.993** | **0.996** | **0.994** | 0.984 | **0.978** |
| I-AUROC | rgb_score | 0.923 | 0.637 | **0.814** | 0.873 | 0.751 | **0.770** | 0.826 | 0.629 | **0.945** | **0.654** | 0.782 |
| | 3D_score | 0.877 | 0.637 | 0.597 | 0.888 | 0.723 | 0.646 | 0.711 | 0.722 | 0.834 | 0.349 | 0.698 |
| | full_score | **0.976** | **0.710** | 0.793 | **0.951** | **0.753** | 0.761 | **0.834** | **0.781** | 0.918 | 0.551 | **0.803** |

These results suggest three important observations. First, each modality individually provides useful anomaly cues, yet neither alone is sufficient for robust detection, as both single-modality variants underperform the full model on average. Second, combining the two modalities through multiplicative agreement suppresses modality-specific false positives and better highlights regions where both modalities indicate inconsistency.

Third, we observe that using the RGB branch alone can outperform the 3D-only variant in most cases and for several categories, occasionally approaching the full score. This is consistent with the design of our replacement module: reconstructed RGB features are refined using 3D guidance through key-value paired retrieval. As a result, the RGB anomaly map is not a purely unimodal signal, but a higher-fidelity representation that already incorporates structural information from the 3D modality. This observation further verifies that the proposed replacement mechanism effectively transfers complementary crossmodal knowledge into the reconstructed RGB space. From a practical perspective, this also suggests a lightweight deployment variant. In resource-constrained settings, or when 3D sensing is noisy or unstable, one may perform inference using only the RGB branch after multimodal training, while still retaining part of the benefits brought by crossmodal learning. Nevertheless, the full model remains the most reliable choice overall.

In conclusion, these findings support our claim that CFR improves anomaly detection by resolving crossmodal ambiguity through complementary evidence, rather than by introducing additional model complexity alone.

# E. Analysis of the PRO Curve

The Area Under the Per-Region Overlap (AUPRO) measures how well predicted anomaly regions overlap with ground-truth defective regions while progressively allowing larger false positive rates (FPR). Compared with conventional pixel-level AUROC, AUPRO better reflects region-level localization quality and is widely adopted in industrial anomaly detection. In practice, reporting AUPRO at different FPR budgets is particularly meaningful, since real deployment often requires operating under strict false-alarm constraints. Lower FPR emphasizes conservative detection behavior, whereas higher FPR reveals the maximal achievable region coverage.

To provide a more complete evaluation beyond the commonly reported AUPRO@30%, we plot the full AUPRO–FPR curves on both MVTec 3D-AD and Eyecandies under the 4-shot setting, as shown in Figure 5. Across both datasets, AUPRO rises rapidly at low FPR and then gradually saturates as more false positives are permitted. This trend indicates that our method can already recover most anomalous regions under relatively strict false-positive constraints, while additional tolerance mainly brings marginal improvements.

For MVTec 3D-AD, the curves are consistently high and tightly clustered across categories. Most classes exceed strong localization performance within a small FPR range and remain stable afterward, suggesting that CFR generalizes reliably across diverse real industrial objects. Only a few categories show slightly slower growth, but all converge to high final scores.

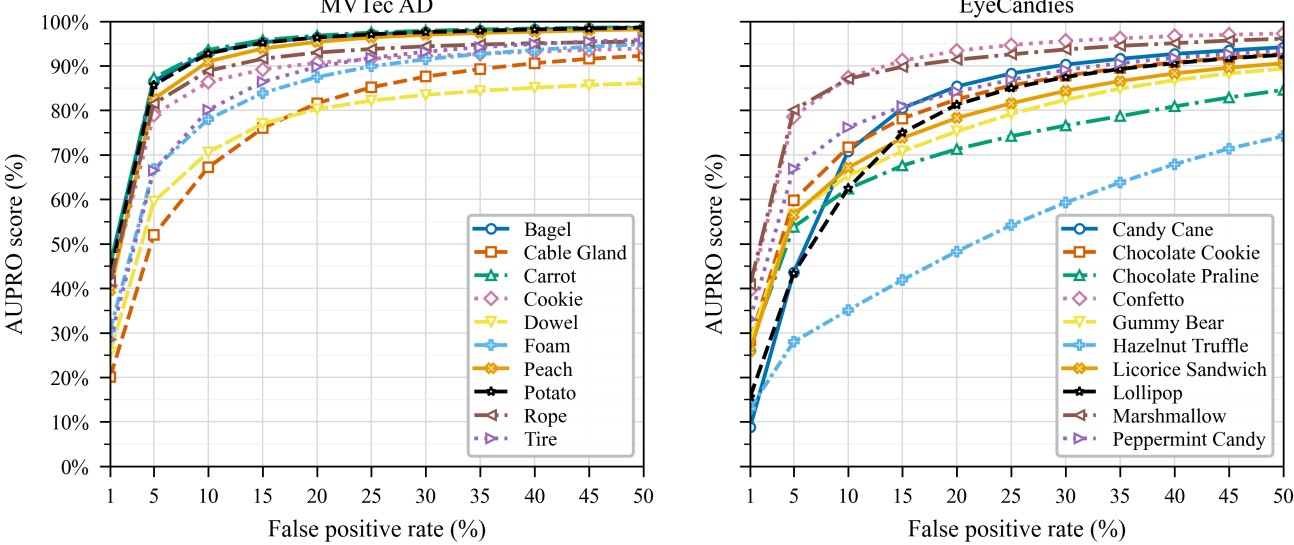

*Figure 5.* AUPRO as a function of the allowed false positive rate (FPR) on all categories of MVTec 3D-AD (left) and Eyecandies (right). Each curve corresponds to one object class, illustrating category-wise localization performance under different operating thresholds.

In contrast, Eyecandies exhibits noticeably larger inter-class variation. Several categories achieve high AUPRO even at very low FPR, while more challenging classes improve gradually and only approach saturation at higher false positive budgets. This behavior is consistent with the stronger appearance diversity, richer texture ambiguity, and larger category-specific difficulty of the synthetic benchmark. Specifically, our method performs poorly on *Hazelnut Truffle*, which we further discuss in Section G. Nevertheless, most classes still converge to high final AUPRO values, demonstrating that our framework remains robust even under substantial cross-category variation.

Overall, these results suggest that CFR not only performs well at the standard AUPRO@30% operating point, but also maintains favorable localization quality across a wide range of deployment requirements, from strict low-false-alarm settings to more permissive scenarios.

## F. Memory Bank Coreset Selection Methods

*Table 12.* Class-wise comparison between CFR 4-shot results with PatchCore-style coreset sampling and random sampling on MVTec 3D-AD. The better value in each pair is highlighted in bold.

| Class | AUPRO@30% | | AUPRO@10% | | AUPRO@5% | | AUPRO@1% | | P-AUROC | | I-AUROC | |
|---|---|---|---|---|---|---|---|---|---|---|---|---|
| | PatchCore | Random | PatchCore | Random | PatchCore | Random | PatchCore | Random | PatchCore | Random | PatchCore | Random |
| bagel | **0.977** | 0.976 | **0.934** | 0.930 | **0.872** | 0.866 | **0.462** | 0.454 | **0.995** | 0.994 | **0.976** | 0.966 |
| cable gland | **0.881** | **0.881** | 0.681 | **0.685** | 0.534 | **0.535** | 0.209 | 0.204 | **0.965** | 0.964 | **0.710** | 0.707 |
| carrot | **0.980** | 0.979 | **0.939** | 0.936 | **0.878** | 0.873 | **0.459** | 0.452 | **0.996** | 0.996 | 0.793 | **0.795** |
| cookie | **0.928** | 0.922 | **0.874** | 0.864 | **0.807** | 0.790 | **0.436** | 0.406 | **0.947** | 0.941 | **0.951** | 0.907 |
| dowel | **0.859** | 0.835 | 0.704 | **0.707** | 0.577 | **0.595** | 0.243 | **0.254** | **0.930** | 0.911 | 0.753 | **0.761** |
| foam | **0.921** | 0.915 | **0.796** | 0.780 | **0.688** | 0.668 | **0.321** | 0.307 | **0.980** | 0.978 | **0.761** | 0.756 |
| peach | **0.971** | 0.970 | **0.913** | 0.910 | **0.831** | 0.827 | **0.398** | 0.395 | **0.993** | 0.993 | 0.834 | **0.851** |
| potato | **0.977** | 0.976 | **0.930** | 0.928 | **0.861** | 0.857 | **0.442** | 0.436 | **0.996** | 0.996 | **0.781** | 0.748 |
| rope | **0.954** | 0.944 | **0.902** | 0.889 | **0.831** | 0.815 | **0.428** | 0.415 | **0.994** | 0.992 | **0.918** | 0.918 |
| tire | **0.939** | 0.932 | **0.820** | 0.801 | **0.691** | 0.665 | **0.305** | 0.285 | **0.984** | 0.981 | 0.551 | **0.555** |
| Average | **0.939** | 0.933 | **0.849** | 0.843 | **0.757** | 0.749 | **0.370** | 0.361 | **0.978** | 0.975 | **0.803** | 0.796 |

Efficiency Analysis in Section B shows that the majority of CFR's additional inference cost arises from nearest-neighbor search over the dense feature repository stored in the memory bank, making memory-bank compression essential. In the main paper, we adopt the same coreset sampling strategy as (Roth et al., 2022) with a sampling ratio of 10%. Specifically, all features collected during few-shot training are clustered, and only the most representative 10% are retained as prototypes. For our proposed key-value 3D-RGB memory bank, we perform downsampling in the key space rather than the value space. This design preserves a richer and more comprehensive value space while maintaining a compact retrieval space, allowing

*Table 13.* Class-wise comparison between PatchCore 4-shot results and random sampling results on Eyecandies. The better value in each pair is highlighted in bold.

| Class | AUPRO@30% | | AUPRO@10% | | AUPRO@5% | | AUPRO@1% | | P-AUROC | | I-AUROC | |
|---|---|---|---|---|---|---|---|---|---|---|---|---|
| | PatchCore | Random | PatchCore | Random | PatchCore | Random | PatchCore | Random | PatchCore | Random | PatchCore | Random |
| CandyCane | **0.904** | 0.901 | **0.711** | 0.702 | **0.438** | 0.418 | 0.077 | **0.083** | **0.969** | **0.969** | 0.549 | **0.570** |
| ChocolateCookie | **0.880** | 0.876 | **0.729** | 0.701 | **0.613** | 0.581 | **0.298** | 0.271 | 0.976 | **0.978** | **0.789** | 0.763 |
| ChocolatePraline | **0.777** | 0.775 | **0.654** | 0.651 | **0.572** | **0.572** | **0.293** | 0.287 | 0.939 | **0.943** | 0.851 | **0.906** |
| Confetto | **0.954** | 0.945 | **0.866** | 0.849 | **0.769** | 0.754 | 0.374 | **0.377** | **0.990** | 0.987 | 0.914 | **0.925** |
| GummyBear | **0.828** | 0.827 | **0.654** | 0.644 | **0.556** | 0.536 | **0.273** | 0.241 | 0.926 | **0.940** | 0.752 | **0.785** |
| HazelnutTruffle | **0.626** | 0.617 | **0.392** | 0.379 | **0.310** | 0.292 | **0.147** | 0.133 | 0.936 | **0.938** | 0.715 | **0.731** |
| LicoriceSandwich | **0.815** | 0.809 | **0.634** | 0.621 | **0.540** | 0.522 | **0.240** | 0.234 | **0.959** | **0.959** | **0.854** | 0.821 |
| Lollipop | **0.864** | 0.862 | **0.591** | 0.587 | 0.389 | **0.391** | 0.111 | **0.114** | **0.970** | 0.965 | **0.651** | 0.634 |
| Marshmallow | 0.940 | **0.943** | **0.879** | 0.872 | **0.816** | 0.802 | **0.433** | 0.418 | 0.993 | **0.994** | **0.909** | **0.909** |
| PeppermintCandy | **0.882** | 0.878 | **0.713** | 0.711 | 0.608 | **0.619** | 0.286 | **0.305** | **0.978** | 0.971 | 0.810 | **0.821** |
| Average | **0.847** | 0.843 | **0.682** | 0.672 | **0.561** | 0.549 | **0.253** | 0.246 | **0.964** | **0.964** | 0.779 | **0.786** |

the attention module to aggregate multiple retrieved candidates and avoiding brittle hard assignment to a single nearest feature.

Table 12 and Table 13 report the full comparison between CFR with coreset sampling and random sampling. As shown, coreset sampling performs better in the majority of pixel-level anomaly localization cases. Moreover, when averaged across all classes, it consistently achieves slightly better or comparable overall performance across the reported metrics. Since the choice of downsampling strategy affects only training-time preprocessing rather than inference latency, coreset sampling provides a favorable trade-off in practice. However, for image-level detection (I-AUROC) on the Eyecandies dataset, coreset sampling does not consistently outperform random sampling. A possible explanation is that the dataset contains abundant yet subtle feature variations, where aggressive clustering may reduce representational diversity. At the pixel level, the model can still retrieve sufficiently similar matches, albeit with slightly larger distances. In contrast, at the image level, the accumulation of many "good but imprecise" matches may cause normal variations to be interpreted as anomalous patterns more frequently.

## G. Failure Case Analysis and Limitations

From our experiments, we identify two primary types of failure cases: (i) *weak ambiguity* and (ii) *unreliable retrieval*. Weak ambiguity refers to scenarios where training samples exhibit limited variation in both appearance and geometry, while unreliable retrieval arises when the underlying data distribution is overly complex or insufficiently covered by the memory bank. In both cases, the benefit of feature replacement becomes marginal, and the model tends to degrade toward a vanilla cyclic reconstruction behavior.

**(i) Weak Ambiguity.** In this regime, the memory bank provides near-complete coverage of local appearances, and the crossmodal mapping becomes close to one-to-one. As a result, the feature distribution is relatively simple and concentrated, allowing the mapping module to fit it almost perfectly. Meanwhile, reconstructed features remain within a narrow and consistent distribution that is well represented by the memory bank. Consequently, only a small portion of features are identified as unreliable, leaving limited opportunity for the replacement module to further refine the reconstruction. In this case, the proposed method effectively reduces to cyclic reconstruction with minimal contribution from retrieval and replacement.

**(ii) Unreliable Retrieval.** In contrast, when appearance variation is excessively large, the crossmodal correspondence becomes difficult to model accurately. Due to limited training samples and memory bank capacity (e.g., downsampling), the retrieved prototypes may be noisy or unrepresentative. Under such conditions, both the retrieval process and the replacement criterion become less reliable. Instead of improving reconstruction fidelity, the replacement step may introduce additional noise, leading to performance degradation.

**Summary of Failure Cases.** These two failure modes reflect different forms of limited discriminability: weak ambiguity corresponds to an overly simple data distribution that is easy to fit, while unreliable retrieval corresponds to a complex distribution that the model fails to capture effectively. The *Hazelnut Truffle* example in Figure 8 exhibits characteristics of both cases. Its appearance shows limited color variation but strong specular and reflection effects, while its geometry contains irregular structures. This combination allows the model to overfit the 3D distribution, yet struggle to establish accurate mappings back to RGB, resulting in suboptimal performance.

**Limitations.** Feature replacement is designed to improve few-shot reconstruction fidelity with limited additional computation, while better leveraging the available normal samples through the inductive bias introduced by filtering, matching, and replacement. CFR learns crossmodal correspondences from few-shot normal data, and the replacement module assists the reconstruction model when such pattern-to-pattern learning remains insufficient. A natural limitation is that the relative benefit of replacement may decrease as the reconstruction problem becomes easier. For example, collecting more normal samples can improve the coverage of valid appearance–geometry correspondences and alleviate crossmodal ambiguity. Similarly, scaling up the reconstruction model can increase its capacity to capture more intricate correspondence patterns. In both cases, the reconstructed features become more reliable, and fewer low-confidence features need to be corrected by the replacement module. Nevertheless, these alternatives are not always practical. Additional normal samples can be costly or difficult to obtain in industrial inspection scenarios, especially for specialized products or rare operating conditions. Increasing model capacity also introduces higher training and inference costs. Therefore, CFR is particularly useful in the practical few-shot setting, where data are limited and a lightweight replacement mechanism offers an efficient way to compensate for imperfect reconstruction.

## H. More Visualization Results

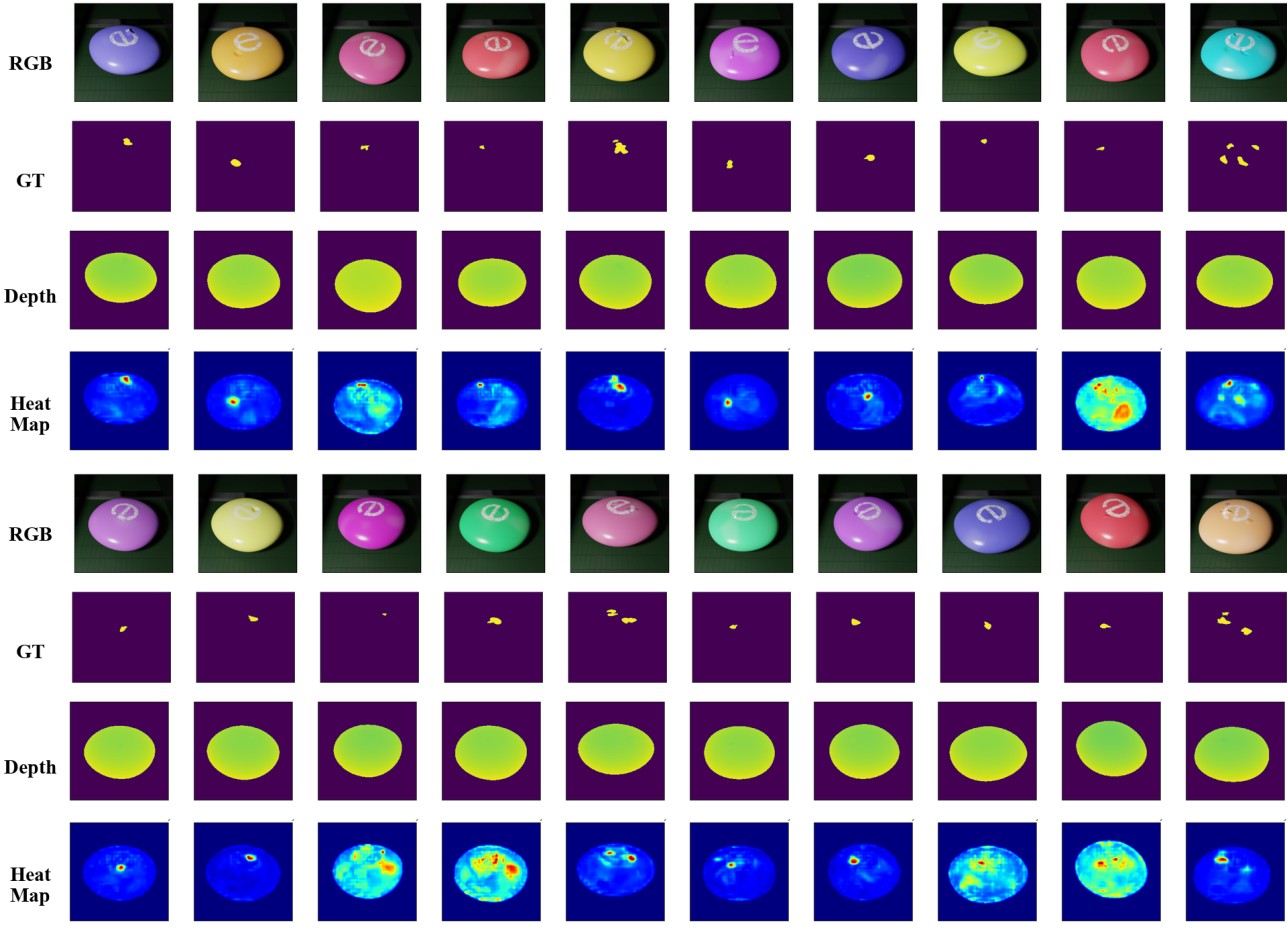

*Figure 6.* Additional qualitative results on **Eyecandies (Confetto)**. Each column shows *RGB*, *Ground-truth*, *Depth*, and the predicted anomaly heatmap. Despite substantial appearance variations, CFR consistently localizes anomalies while avoiding reconstruction collapse.

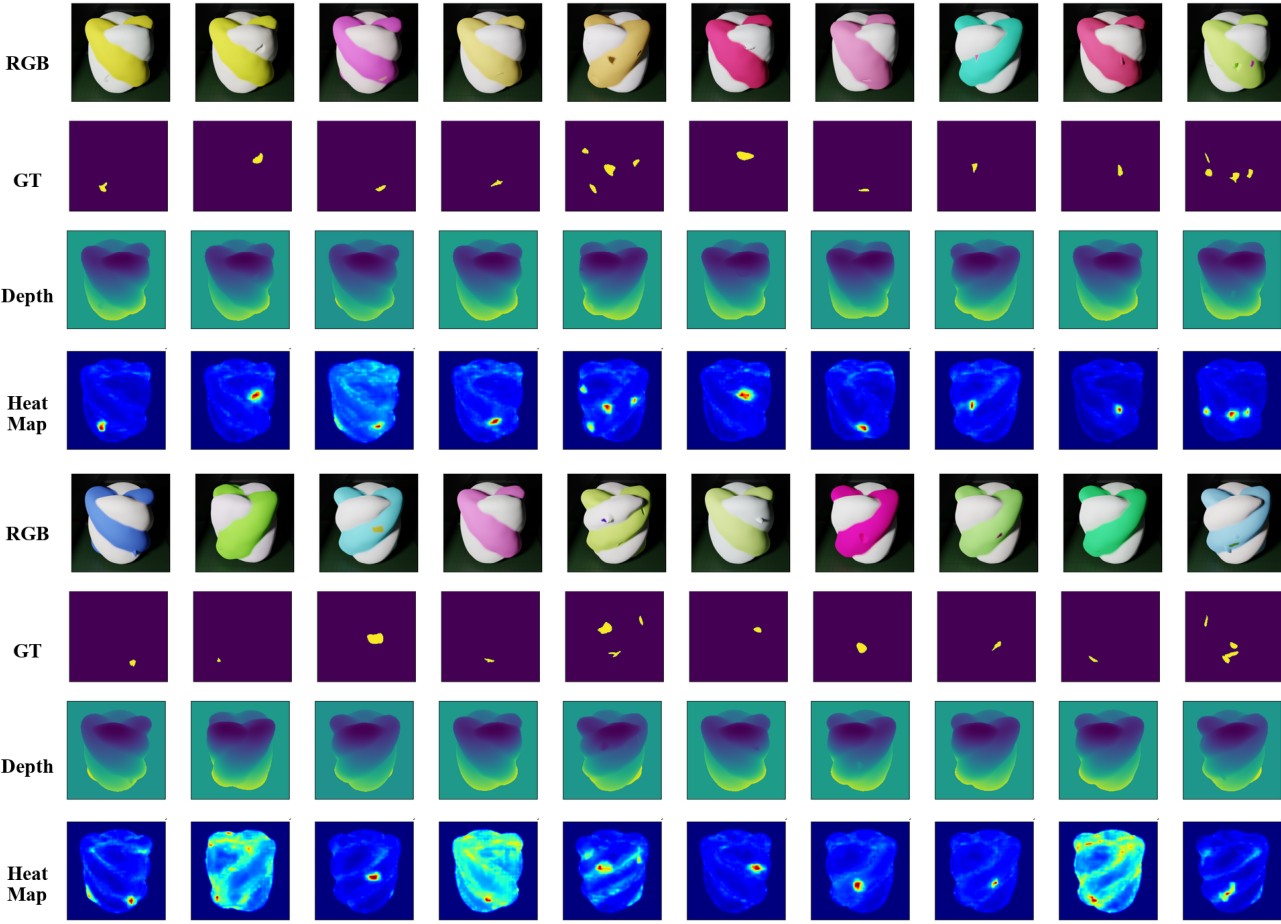

*Figure 7.* Additional qualitative results on **Eyecandies (Marshmallow)**. Each column shows *RGB*, *Ground-truth*, *Depth*, and the predicted anomaly heatmap. Despite substantial appearance variations, CFR consistently localizes anomalies while avoiding reconstruction collapse.

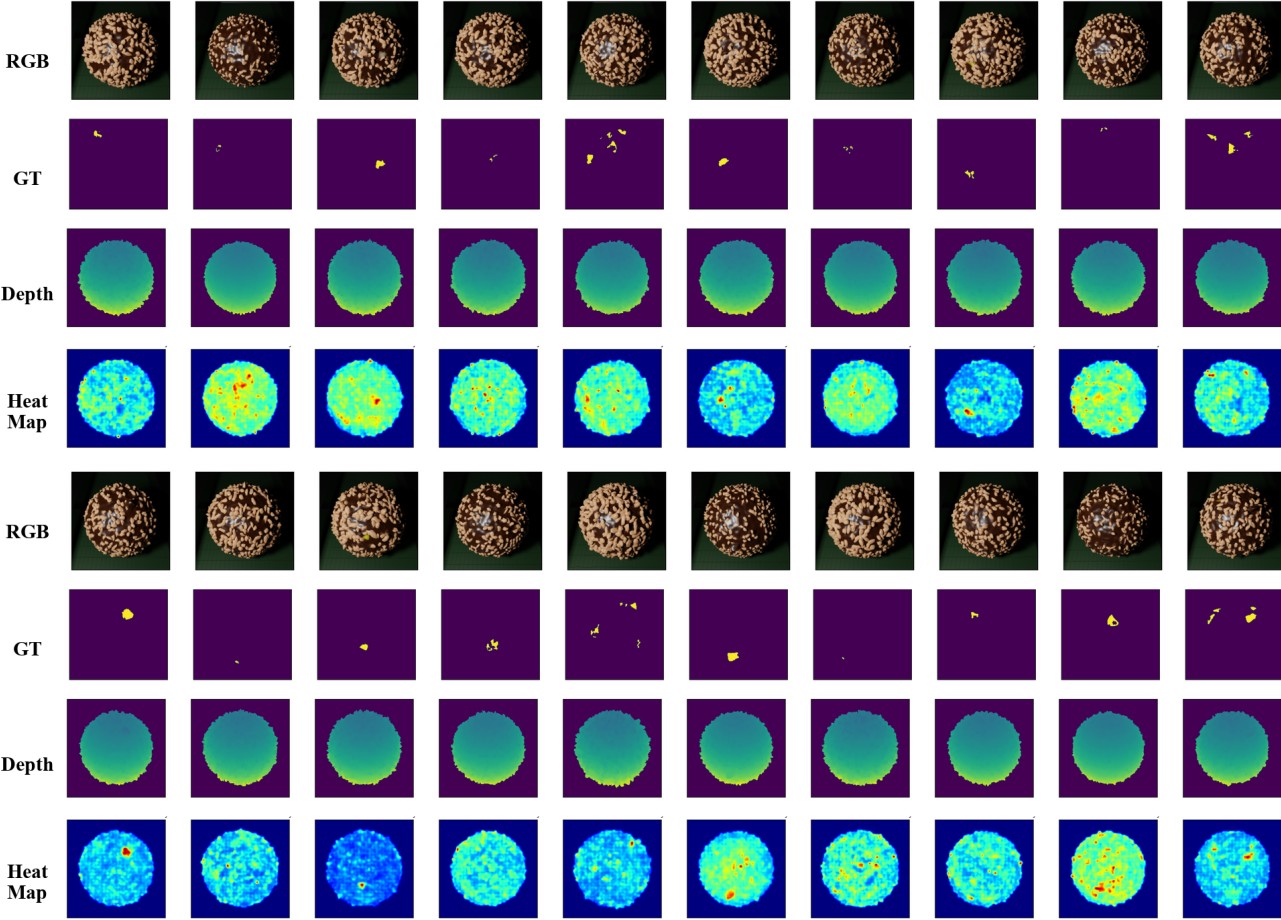

*Figure 8.* Additional qualitative results on **Eyecandies (Hazelnut Truffle)**. Each column shows *RGB*, *Ground-truth*, *Depth*, and the predicted anomaly heatmap.

