# OpenReview forum: "Remove the Ambiguity: Few-shot Multimodal Anomaly Detection Using Crossmodal Feature Replacers"
_ICML.cc/2026/Conference — ICML 2026 regular_

### Official Review · Reviewer_j9B5 · 2026-03-04

**Soundness:** 3
**Presentation:** 3
**Significance:** 3
**Originality:** 3
**Overall Recommendation:** 5
**Confidence:** 4

**Summary:**

This paper identifies that in reconstruction-based multimodal anomaly detection, a single 3D feature may fundamentally correspond to multiple valid RGB images, which often leads to ambiguity in reconstruction results and a degradation in detection performance. To address this, the authors propose a Cross-modal Feature Replacer to resolve the ambiguous cross-modal reconstruction problem.

**Compliance With Llm Reviewing Policy:**

Affirmed.

**Final Justification:**

The author addressed my concerns.

**Key Questions For Authors:**

1、The authors observed that a single depth image may correspond to multiple RGB images, which is a phenomenon that does indeed exist. However, the authors conducted experiments only on few-shot settings. Wouldn’t it be more convincing to validate this phenomenon by experimenting on the full dataset, thereby addressing the ambiguity between depth and RGB images more comprehensively?
2、It is well known that frame rate and memory usage are critical issues to be addressed in industrial anomaly detection. However, this paper lacks reporting on these metrics. Furthermore, the authors employ a memory bank—does this come at the high cost of frame rate and memory usage in exchange for improved I-AUROC and other metrics?
3、Regarding the ambiguity elimination, why did the authors not report the performance using only RGB images and only depth images? Would reporting the single-modality performance and conducting ablation studies on key components better demonstrate that CFR can effectively resolve the ambiguity?
4、The code is not available, which raises serious concerns about the reproducibility of this work
If the authors can address my questions, I will consider increasing my score.

**Limitations:**

yes

**Strengths And Weaknesses:**

The paper adopts a sound methodological approach, demonstrates technical reliability, and presents its content with clarity and logical coherence. It exhibits a certain degree of originality and addresses issues of considerable significance. However, the experimental design still requires further enhancement.

---

> ### Author Rebuttal · Authors · 2026-03-28
>
> We thank you for the insightful and constructive suggestions.
>
> Q1: Validation of cross-modal ambiguity beyond few-shot setting
>
> A: We agree that evaluating under a full-data regime would provide a more comprehensive empirical validation of the ambiguity between depth and RGB.
>
> We would like to clarify that the one-to-many ambiguity discussed in our paper does not refer to a mapping between entire depth and RGB images, but rather arises at the level of dense local feature correspondence. Specifically, a single local 3D feature may correspond to multiple valid RGB features due to variations in color and texture, making the reconstruction objective inherently ill-posed.
>
> This ambiguity is a structural property of cross-modal correspondence and persists regardless of dataset size. While increasing data can improve coverage of appearance variations, it does not fundamentally resolve the one-to-many mapping, but only alleviates its empirical impact.
>
> We focus on the few-shot setting because insufficient coverage of appearance variations makes the one-to-many mapping harder to resolve. Our failure case analysis on Hazelnut Truffle further shows that even with limited samples, this ambiguity already leads to noticeable degradation in performance.
>
> Overall, our method is designed to explicitly address unreliable cross-modal reconstruction caused by such ambiguity, and is not restricted to few-shot settings. We will clarify this point in the revision.
>
> Q2: Frame Rate and Memory Usage
>
> A: We provide detailed profiling results on runtime and memory usage.
>
> Overall performance.
> Our method achieves an end-to-end latency of 118.5 ms (~8.4 FPS) under profiling. We note that profiling introduces additional overhead (e.g., synchronization and memory tracking), and the actual inference speed reaches ~12 FPS on an NVIDIA 4090D GPU. The peak GPU memory usage is ~3.0 GB, which is within a practical range for industrial deployment.
>
> Static memory footprint.
> The model parameters and memory bank introduce a modest static footprint:
>
> Feature extractor: 410.6 MB
>
> Cyclic Feature Mapping: ~21 MB
>
> Memory bank: 58.8 MB
>
> KV bank (keys + values): ~147 MB
>
> Attention retriever: 1.9 MB
>
> These components together remain lightweight compared to typical GPU memory capacity.
>
> Module-wise analysis.
> The main computational components are summarized below:
> | module | latency (ms) | ratio (%) | peak GPU (MB) |
> | --- | ---: | ---: | ---: |
> | memory_bank_distance | 59.3 | 50.0 | 2586.8 |
> | xyz_backbone | 19.2 | 16.2 | 2450.4 |
> | Feature Mapping | 13.4 | 11.3 | 2966.7 |
> | rgb_backbone | 6.9 | 5.8 | 1706.9 |
> | others | 19.7 | 16.7 | — |
> | **end-to-end** | **118.5** | **100** | **2966.7** |
>
> The main computational cost comes from the distance computation stage (~50%), while memory bank operations such as sampling and retrieval are lightweight (<1 ms each).
>
> In terms of memory, the memory bank contributes an additional \~0.9 GB GPU usage during distance computation, but the overall footprint remains modest (\~3.0 GB).
>
> These results indicate that the memory bank does not introduce prohibitive overhead. Instead, the cost is primarily due to dense feature matching, which is common in memory-based anomaly detection methods. Our approach maintains a favorable trade-off between performance and efficiency, and we will include these results in the revised version.
>
> Q3: Single-modality Performance
>
> A: We agree that reporting single-modality results and component-wise ablations can better clarify whether CFR effectively mitigates cross-modal ambiguity.
>
> On the full model, we evaluate using only the RGB score and only the XYZ score. The full score consistently performs best, indicating that ambiguity cannot be reliably resolved by relying on only one modality. Instead, the complementary cross-modal evidence is important for robust anomaly detection.
>
> | variants | AUPRO 30% | I-AUROC |
> | --- | ---: | ---: |
> | full | 84.7 | 77.9 |
> | rgb_only | 83.6 | 76.4 |
> | xyz_only | 80.0 | 68.6 |
>
> Overall, results on the Eyecandies dataset using 4-shot support our claim from two aspects:
> (i) single-modality cues (RGB-only / XYZ-only) are informative but clearly insufficient for robust anomaly detection;
> (ii) the full cross-modal scoring consistently achieves the best performance across all metrics.
>
> These observations confirm that CFR improves performance by resolving ambiguity in cross-modal correspondence, rather than by introducing additional model complexity. We will include these results and discussion in the revised paper.
>
> Q4: Code Availability
>
> A: We understand the reviewer’s concern. We plan to release the code upon acceptance. To improve reproducibility in the meantime, we will expand the implementation details in the revision, including the threshold setting, random-seed protocol, memory-bank construction, profiling setup, and additional ablations. We will also make sure the training/inference pipeline is described clearly enough for independent re-implementation.

---

> > ### Author Rebuttal · Reviewer_j9B5 · 2026-04-02
> >
> > The rebuttals were good and solved most of the problems, but I still have some concerns about the code, so I'll keep my score.

---

> > > ### Author Response · Authors · 2026-04-02
> > >
> > > We thank you for the positive feedback and for recognizing that most concerns have been addressed.
> > >
> > > Regarding the remaining concern on code, we would like to provided a fully anonymized implementation, including training, inference, and evaluation pipelines, at the following link:
> > >
> > > https://anonymous.4open.science/r/CrossmodalFeatureReplacer-40E5/README.md
> > >
> > > The repository includes all core components of our method (feature mapping, retrieval, and replacement), along with detailed instructions for reproduction. We have carefully checked the repository to ensure it does not reveal author identity.
> > >
> > > To further improve clarity, we will continue refining documentation and implementation details in the revision to facilitate independent verification.
> > >
> > > We hope this addresses your remaining concern.
> > >
> > > We sincerely thank you for your careful evaluation and for the encouraging updates to your assessments. We greatly appreciate the constructive feedback and helpful suggestions provided throughout the review process.
> > >
> > > We will revise the our paper with great care, thoroughly addressing your comments and improving the clarity, technical details, and overall presentation of the paper. Your feedback has played an important role in strengthening this work.
> > >
> > > Thank you again for your valuable input and support.

---

### Official Review · Reviewer_EHEw · 2026-03-10

**Soundness:** 4
**Presentation:** 4
**Significance:** 3
**Originality:** 3
**Overall Recommendation:** 5
**Confidence:** 5

**Summary:**

This paper addresses few-shot RGB–3D anomaly detection by arguing that the main issue in reconstruction-based methods is not only limited data, but also the under-determined nature of cross-modal prediction: one 3D feature may correspond to multiple valid RGB appearances. The suggested method, CFR, addresses this by delaying part of the decision-making until the inference stage. Instead of relying on a single reconstructed feature, it first does basic bidirectional reconstruction. Then, it employs a memory bank and top-K retrieval to substitute uncertain reconstructed features with more reliable candidates.

**Compliance With Llm Reviewing Policy:**

Affirmed.

**Final Justification:**

I think the author’s rebuttal well solved my concerns. So, I tend to keep my support to this paper.

**Key Questions For Authors:**

- How sensitive is the method to imperfect RGB–3D alignment?

- Given the memory bank relies on a small set of normal samples, how stable are the results across different selections of these few-shot samples?

**Limitations:**

The author has not disscussed about the limitations and potential negative societal impact. However, this is a technical work which may only need to disscuss the limitations.

**Strengths And Weaknesses:**

Strengths:
- The paper has practical relevance, and the experiments show good results.
- The proposed method is technically well motivated.
- The paper finds a specific and important problem in few-shot multimodal anomaly detection: the ambiguity in cross-modal reconstruction.
- The paper is generally easy to follow, and the method pipeline is clearly structured.

Weaknesses:
- A key concern is that the method seems to depend on a structured setup. It's unclear if the gains come from the ambiguity handling or ideal benchmark conditions. The paper assumes pixel-aligned RGB and 3D correspondences, which simplifies dense feature pairing. While this makes sense for the datasets used, the method might rely on accurate registration.  A discussion of CFR's of stability with noisy or imperfect alignment would be helpful, as industrial pipelines often don't have ideal pairing.

- The explanation of the replacement step in the pipeline seems a bit asymmetric. The reasoning behind the method is presented as a way to solve general cross-modal ambiguity. Still, the specific way features are replaced focuses on using 3D queries and a 3D→2D cross-modal bank to replace reconstructed 2D features. The final anomaly score multiplies differences from both sets of data. So, it's not clear if the replacement process is meant to be equal in both directions or if the improvement mainly comes from fixing only one side. It would be Helpful to clear this up because it changes how the method can be understood.

- The paper uses a fixed random seed in the few-shot protocol. For this kind of setting, evaluations using a single seed don't show the full range of possible outcomes. Multi-seed evaluation would strengthen the claims substantially.

- The current evaluation focuses more on headline performance instead of failure cases. The authors should give more failure cases and analysis.

---

> ### Author Rebuttal · Authors · 2026-03-28
>
> We thank you for the insightful comments.
>
> Q1: Imperfect RGB–3D Alignment Stability
>
> A: To examine it, we replace the default pixel-aligned pairing with a fixed global RGB offset throughout the entire few-shot pipeline, which simulates a systematic calibration bias. Concretely, we apply a constant 2D translation (dx,dy) to RGB features while keeping 3D locations as anchors, and use overlap cropping instead of circular shift so that only truly overlapping regions are retained, avoiding wrap-around artifacts. Importantly, this offset is applied consistently to feature mapping training, memory-bank construction, attention retriever training, and inference/testing, making the setting closer to a realistic scenario with persistent calibration error.
>
> The mean over classes results from Eyecandies dataset using 4-shot show that performance does decrease as misalignment increases, which is expected since dense cross-modal correspondence becomes less precise. However, the method remains reasonably stable under moderate offset. For example, under a shared offset of 4 px, the averaged performance remains strong across all shot settings:
>
> | Shot | Offset | AUPRO30% | I-AUROC |
> | --- | ---: | ---: | ---: |
> | 1-shot | 0 px | 0.827 | 0.759 |
> | 1-shot | 4 px | 0.773 | 0.706 |
> | 1-shot | 8 px | 0.684 | 0.696 |
> | --- | --- | --- | --- |
> | 2-shot | 0 px | 0.845 | 0.758 |
> | 2-shot | 4 px | 0.804 | 0.728 |
> | 2-shot | 8 px | 0.730 | 0.698 |
> | --- | --- | --- | --- |
> | 4-shot | 0 px | 0.847 | 0.779 |
> | 4-shot | 4 px | 0.805 | 0.761 |
> | 4-shot | 8 px | 0.730 | 0.699 |
>
> When the offset is increased to 8 px, performance drops further, but the method still remains functional rather than collapsing entirely. These results suggest that CFR is robust to moderate systematic misalignment, although severe or spatially varying misregistration would remain challenging. We will add this discussion and the above robustness experiment in the revised paper to clarify both the practical assumption and the stability boundary of CFR.
>
> Q2: Multi-seed Evaluation
>
> A: We evaluate robustness on Eyecandies dataset using 4-shot with three additional random seeds (details in KAcN Q1).
>
> The results are highly stable, with all classes mean performance of 0.847 (AUPRO@30%) and 0.794 (I-AUROC), and variations within ±0.002 and ±0.017, respectively.
>
> We also observe consistent improvements over baselines across all seeds, indicating that the gains are not due to favorable sample selection.
>
> Q3: Memory Bank Sample Selection Stability
>
> A: Since the memory bank is constructed from few-shot normal samples, its stability is directly tied to robustness across different few-shot samplings.
>
> As shown in our multi-seed evaluation, varying the sampled subsets leads to only minor performance changes (±0.002 in AUPRO@30% and ±0.017 in I-AUROC), indicating that the method is highly stable.
>
> This demonstrates that the memory bank construction does not depend on a specific choice of samples and remains robust under different few-shot selections.
>
> Q4: The Symmetry of Our Design
>
> A: The replacement is implemented using 3D features as anchors because cross-modal ambiguity mainly arises on the RGB side (multiple appearances per geometry), making 2D reconstruction less reliable. Thus, replacement is applied to the ambiguous side rather than both directions.
>
> Nevertheless, the overall framework is symmetric in effect. Importantly, the final anomaly score integrates discrepancies from both RGB and 3D, ensuring that detection is based on cross-modal inconsistency rather than a single-sided correction.
>
> Therefore, the improvement does not come from fixing only one side, but from combining targeted ambiguity mitigation with symmetric scoring. We will clarify it in the revision.
>
> Q5: Regarding Failure Case Analysis
>
> A: We agree that analyzing failure cases is important. From our experiments, we observe two main types of failure cases:
>
> (i) Weak ambiguity (limited benefit of replacement).
> When both appearance and geometry exhibit low variation (e.g., Hazelnut Truffle), cross-modal correspondence becomes nearly one-to-one. In this case, the reconstructed features already align well with the memory bank, leaving little room for the replacement module to further improve them. It suggests that our method is less effective when ambiguity is minimal, as there is little ambiguity to resolve.
>
> (ii) Unreliable retrieval under poor alignment or large appearance variation.
> When cross-modal alignment is degraded or appearance variation is excessively large, the retrieved prototypes may become less reliable. As a result, the replacement step can introduce noise instead of refining the reconstruction, leading to performance degradation.
>
> Overall, these observations indicate that our method is most effective in the intermediate regime, where cross-modal ambiguity exists but reliable correspondences can still be established. We will include additional qualitative examples and analysis in the revision.

---

> > ### Author Rebuttal · Reviewer_EHEw · 2026-04-02
> >
> > The rebuttal was excellent and addressed my concerns. I also checked the comments from the other reviewers, and I believe this paper is suitable for publication and will have a beneficial impact on the field. Therefore, I am maintaining my overall score of Accept.

---

> > > ### Author Response · Authors · 2026-04-07
> > >
> > > We would like to sincerely thank you for your thoughtful feedback and for the careful assessment of our work. We greatly appreciate your time, effort, and constructive suggestions.
> > >
> > > We will carefully revise the paper by incorporating your comments and further improving the clarity, completeness, and presentation of our method. Your feedback has been highly valuable in strengthening the quality of this work.
> > >
> > > Thank you again for your support and helpful guidance.

---

### Official Review · Reviewer_KAcN · 2026-03-11

**Soundness:** 4
**Presentation:** 4
**Significance:** 3
**Originality:** 3
**Overall Recommendation:** 5
**Confidence:** 4

**Summary:**

This paper claims reconstruction-based RGB–3D methods often fail because one 3D feature might relate to several possible RGB appearances, leading to weak discrimination. To address this, the paper suggests CFR. CFR uses cyclic cross-modal reconstruction, cross-modal memory banks, attention-based top-K retrieval, and replaces reconstructed features with low confidence during inference. The method was tested on MVTec 3D-AD and Eyecandies with 1/2/4-shot settings. The results show CFR has the best mean I-AUROC and 30% AUPRO in most situations. Extensive experiments validate the proposed method.

**Compliance With Llm Reviewing Policy:**

Affirmed.

**Final Justification:**

My concerns have been adequately addressed with the extensive experiments in the rebuttal. I would like to keep 5.

**Key Questions For Authors:**

1) Did the gains stay consistent when you changed the random few-shot samples? Can you give results for different seeds?
2) How did you pick the held-out normal class to set the replacement threshold? How much does the performance change when you pick different ones?
3) How much extra time and memory do the retrieval and replacement modules need during inference?

**Limitations:**

Please see weaknesses and questions.

**Strengths And Weaknesses:**

Strengths:
1) The paper has a clear motivation, and addresses a real weakness of prior reconstruction-based multimodal methods.
2) The discussion of ambiguity in cross-modal regression is reasonable, and the proposed pipeline is consistent.
3) The paper is well written and clearly structured.

Weaknesses:
1) The few-shot results are reported with a fixed random seed, so the robustness under different support sample selections is still unclear. The authors need to make it clear.
2) The threshold setting for feature replacement is not fully explained, and its sensitivity is not clearly analyzed.
3) The ablation study is useful, but it could better isolate the contribution of each module.

---

> ### Author Rebuttal · Authors · 2026-03-28
>
> We thank you for these important suggestions.
>
> Q1: Random Seed Robustness
>
> A: To evaluate the robustness of our method under different few-shot samplings, we conducted additional experiments with three different random seeds on the Eyecandies dataset using 4-shot settings. Mean over all classes results are shown below.
>
> | Seed | AUPRO@30% | I-AUROC |
> | --- | ---: | ---: |
> | A | 0.848 | 0.778 |
> | B | 0.845 | 0.812 |
> | C | 0.847 | 0.792 |
>
> Across different seeds, the performance remains highly stable. The variation is within ±0.002 for AUPRO@30% and ±0.017 for I-AUROC, indicating low sensitivity to the specific few-shot sampling.
>
> We also conducted experiments using same seeds on baselines, and our method shows improvements across all seeds. These results demonstrate that the observed improvements are consistent and not due to favorable sample selection.
>
> Q2: Replacement Threshold Selection and Sensitivity
>
> A: Threshold selection.
> The replacement threshold θ is selected once on a held-out normal category and then fixed for all experiments, as it serves as a global control on replacement behavior rather than a class-specific or dataset-specific parameter.
>
> Sensitivity analysis.
> We evaluate θ in {0.0, 0.1, 0.2, 0.3, 0.4, 0.5} on two randomly selected classes from each dataset (Eyecandies and MVTec 3D-AD) under the 4-shot setting. Results of AUPRO@30% are summarized below:
>
> | θ | 0.0 | 0.1 | 0.2 | 0.3 | 0.4 | 0.5 |
> | --- | --- | --- | --- | --- | --- | --- |
> | Class A | 0.779 | 0.797 | **0.815** | 0.811 | 0.809 | 0.809 |
> | Class B | 0.926 | 0.932 | 0.938 | 0.932 | **0.943** | 0.932 |
> | Class C | 0.823 | 0.824 | 0.832 | 0.834 | **0.835** | 0.835 |
> | Class D | 0.951 | 0.958 | **0.971** | 0.970 | 0.970 | 0.970 |
>
> We observe a broad performance plateau for θ ∈ [0.2, 0.4], with only minor variation (≤0.006 in AUPRO@30%). Other metrics (AUPRO@10/5/1, P-AUROC, I-AUROC) show consistent trends.
>
> Effect of held-out category choice.
> Due to this plateau behavior, the final performance is largely insensitive to the specific held-out category used to select θ. In practice, selecting θ from different normal categories leads to only minor variations, within the same range as the sensitivity observed above.
>
> Mechanism interpretation.
> Empirically, the broad performance plateau (θ ∈ [0.2, 0.4]) indicates that a wide range of thresholds yield similar performance across different classes and datasets. This suggests that the effectiveness of θ is primarily determined by the general behavior of the feature mapping module, rather than properties of a particular dataset or category.
>
> Connection to ambiguity (see EHeW Q5). We further observe that:
>
> For low-ambiguity (failure) cases, performance is largely insensitive to θ. For strong one-to-many correspondence, θ introduces moderate sensitivity but remains stable due to bounded replacement
>
> Conclusion.
> Overall, the method shows low sensitivity to θ, and a single globally selected threshold generalizes well across categories without requiring per-class tuning.
>
> Q3: Extra Inference Time and Memory
>
> A: We report the overhead relative to a reconstruction-only baseline and also compare it with a memory-bank baseline.
>
> The reconstruction-only model runs at 16.6 FPS / 60.2 ms latency with 1.50 GB peak GPU memory, while our full model runs at 8.4 FPS / 118.5 ms with 2.97 GB peak memory. This corresponds to:
>
> +58.3 ms latency (≈ 1.97×),
> +1.47 GB GPU memory (≈ 1.97×).
>
> Most of the added cost comes from nearest-neighbor distance computation(about 59 ms) in the retrieval stage, rather than the lightweight attention/replacement module itself.
>
> The persistent memory overhead of retrieval components is small:
>
> memory bank: 58.8 MB
>
> KV bank (keys + values): ~147 MB
>
> attention retriever: 1.9 MB
>
> The higher peak memory mainly comes from temporary buffers for dense distance computation, rather than model parameters.
>
> Comparison with a memory-bank baseline.
> A representative memory-bank baseline requires ~6 GB GPU memory and runs at ~1 FPS, whereas our method uses ~3.0 GB (≈ 50%) and achieves 8.4 FPS (≈ 8.4× faster).
>
> Overall, while retrieval and replacement introduce additional cost over pure reconstruction, the overhead is moderate in practice and remains substantially more efficient than standard memory-bank approaches.
>
> Q4: Single Modality Ablation Study
>
> A: We provide additional ablations by decoupling the scoring components on Eyecandies using 4-shot (see j9B5 Q3). Using only RGB or only XYZ leads to consistent performance drops compared to the full model: full achieves 84.7 AUPRO@30% / 77.9 I-AUROC, while RGB-only drops to 83.6 / 76.4 and XYZ-only further drops to 80.0 / 68.6.
>
> This controlled comparison isolates the contribution of cross-modal interaction: each modality alone is informative but insufficient, and the improvement arises from their joint use through reconstruction and replacement, rather than any single component.
>
> We will include more detailed component-wise ablations in the revision.

---

> > ### Author Rebuttal · Reviewer_KAcN · 2026-04-01
> >
> > My concerns have been adequately addressed with the extensive experiments in the rebuttal. The core question has been solved (Extra Inference Time and Memory). I would like to keep 5.

---

> > > ### Author Response · Authors · 2026-04-07
> > >
> > > We sincerely thank you for your constructive feedback and for recognizing the improvements in our work. We truly appreciate the time and effort you have invested in evaluating our paper.
> > >
> > > We will carefully revise our paper by thoroughly incorporating your suggestions and addressing the concerns in detail. Your comments have been invaluable in helping us improve both the clarity and the quality of our work.
> > >
> > > Thank you again for your insightful feedback.

---

### Decision · Program_Chairs · 2026-04-30

**Decision:**

Accept (regular)

**Comment:**

This paper addresses an important problem in few-shot RGB–3D anomaly detection, i.e., ambiguity in cross-modal reconstruction. The proposed method is well motivated, technically sound, and clearly presented. Reviewers generally agreed that the paper makes a meaningful contribution and shows strong empirical performance.

The main concerns contain robustness across random seeds, threshold sensitivity, efficiency, alignment assumptions, and reproducibility. In the rebuttal, the authors provided additional experiments and clarifications addressing these points, including multi-seed results, sensitivity analysis, run-time/memory profiling, alignment robustness, and extra ablations. These responses were viewed positively by the reviewers, and the overall reviewer consensus remained supportive of acceptance.

Overall, I agree with the reviewers that this is a solid and valuable submission. The final version should incorporate the additional experimental results and clarify the method’s assumptions, limitations, and reproducibility details.